# *Vicia faba* SV channel VfTPC1 is a hyperexcitable variant of plant vacuole Two Pore Channels

Jinping Lu[1,2], Ingo Dreyer[3], Miles Sasha Dickinson[4], Sabine Panzer[5], Dawid Jaślan[1,6], Carlos Navarro-Retamal[3,7], Dietmar Geiger[1], Ulrich Terpitz[5], Dirk Becker[1], Robert M Stroud[4]*, Irene Marten[1]*, Rainer Hedrich[1]

[1]Julius-Maximilians-Universität (JMU), Biocenter, Department of Molecular Plant Physiology and Biophysics, Würzburg, Germany; [2]School of Life Sciences, Zhengzhou University, Zhengzhou, China; [3]Universidad de Talca, Faculty of Engineering, Center of Bioinformatics, Simulation and Modeling, Talca, Chile; [4]University of California San Francisco, Department of Biochemistry and Biophysics, San Francisco, United States; [5]Julius-Maximilians-Universität (JMU), Biocenter, Theodor-Boveri-Institute, Department of Biotechnology and Biophysics, Würzburg, Germany; [6]Ludwig Maximilians-Universität, Faculty of Medicine, Walther Straub Institute of Pharmacology and Toxicology, Munich, Germany; [7]Department of Cell Biology and Molecular Genetics, University of Maryland, College Park, United States

*For correspondence:
stroud@msg.ucsf.edu (RMS);
irene.marten@uni-wuerzburg.
de (IM)

Competing interest: The authors declare that no competing interests exist.

**Abstract** To fire action-potential-like electrical signals, the vacuole membrane requires the two-pore channel TPC1, formerly called SV channel. The TPC1/SV channel functions as a depolarization-stimulated, non-selective cation channel that is inhibited by luminal $Ca^{2+}$. In our search for species-dependent functional TPC1 channel variants with different luminal $Ca^{2+}$ sensitivity, we found in total three acidic residues present in $Ca^{2+}$ sensor sites 2 and 3 of the $Ca^{2+}$-sensitive AtTPC1 channel from *Arabidopsis thaliana* that were neutral in its *Vicia faba* ortholog and also in those of many other Fabaceae. When expressed in the *Arabidopsis* AtTPC1-loss-of-function background, wild-type VfTPC1 was hypersensitive to vacuole depolarization and only weakly sensitive to blocking luminal $Ca^{2+}$. When AtTPC1 was mutated for these VfTPC1-homologous polymorphic residues, two neutral substitutions in $Ca^{2+}$ sensor site 3 alone were already sufficient for the *Arabidopsis* At-VfTPC1 channel mutant to gain VfTPC1-like voltage and luminal $Ca^{2+}$ sensitivity that together rendered vacuoles hyperexcitable. Thus, natural TPC1 channel variants exist in plant families which may fine-tune vacuole excitability and adapt it to environmental settings of the particular ecological niche.

## Editor's evaluation

Plant intracellular ion channels are poorly understood. In this manuscript, patch-clamp is used to define functional differences between two cation channels present in the vacuole. The authors present valuable findings that indicated a calcium-biding site is responsible for increased excitability in the vacuole of the faba bean plant. The experimental evidence presented is convincing and findings have practical implications for the subfield of plant electrophysiology.

## Introduction

Soon after the patch-clamp technique was first applied to plant cells (for review see *Hedrich and Marten, 2011*; *Schroeder and Hedrich, 1989*), the slow vacuolar (SV) channel was identified as a

calcium-regulated, voltage-dependent cation channel (*Allen and Sanders, 1996*; *Amodeo et al., 1994*; *Bethke and Jones, 1994*; *Colombo et al., 1988*; *Hedrich and Neher, 1987*; *Pottosin et al., 2001*; *Ward and Schroeder, 1994*, for review see *Pottosin and Dobrovinskaya, 2022*; *Ward, 2022*). The SV channel is similarly permeable for $Na^+$ and $K^+$, and in principle also to $Ca^{2+}$. However, the physiological relevance of a potential $Ca^{2+}$ permeability in planta is highly controversial (for review see *Hedrich et al., 2018*; *Pottosin and Dobrovinskaya, 2022*), a view further supported – among other studies – by recent molecular dynamics (MD) simulations (*Navarro-Retamal et al., 2021*). About two decades after the discovery of SV channels, *Peiter et al., 2005* revealed that the *Arabidopsis thaliana* SV channel is encoded by the single copy gene *TPC1*. Since then, the AtTPC1 channel has become a model for understanding the physiological role and function of SV/TPC1 channels in higher plants. Studies on AtTPC1 mutant plants suggest that TPC1 is not responsible for global $Ca^{2+}$ responses but contributes to long-distance electrical and $Ca^{2+}$ signaling under salt stress conditions and is an essential player in vacuole excitability (*Bellandi et al., 2022*; *Choi et al., 2014*; *Evans et al., 2016*; *Jaślan et al., 2019*; *Ranf et al., 2008*, for review see *Hedrich et al., 2018*; *Pottosin and Dobrovinskaya, 2022*).

Within the tree of live, there are TPC1-like sequences in animals and plants (see for review *Jaślan et al., 2020*; *Kintzer and Stroud, 2018*). During land plant evolution characteristic structural fingerprints of TPC1-type channels remained unchanged from mosses to flowering plants (*Dadacz-Narloch et al., 2011*; *Dreyer et al., 2021*; *Hedrich et al., 1988*). Recently, both the crystal and cryoEM structure of the *Arabidopsis* AtTPC1 channel and thus the molecular topology of this vacuolar cation channel became available (*Dickinson et al., 2022*; *Guo et al., 2016*; *Kintzer and Stroud, 2016*; *Ye et al., 2021*). AtTPC1 is formed by a dimer whose monomers consist of two tandem Shaker-like cassettes, each with six transmembrane domains (TM1–6), connected by a cytoplasmic loop with two EF hands. Calcium binding to the EF hands is necessary for SV/TPC1 channel activation at elevated cytosolic $Ca^{2+}$ levels (*Guo et al., 2016*; *Schulze et al., 2011*; *Ye et al., 2021*). Structural 3D motif comparison and point mutation analysis identified the major voltage-sensing domain (VSD) required for depolarization-dependent activation in each monomer in the first four transmembrane segments of the second Shaker-like cassette (*Guo et al., 2016*; *Jaślan et al., 2016*). CryoEM structures further suggested that activation of VSD2 is required for $Ca^{2+}$ activation of the EF hand domain (*Ye et al., 2021*). In contrast, channel activation is strongly suppressed when the $Ca^{2+}$ level in the vacuolar lumen reaches 1 mM and more (*Guo et al., 2016*; *Jaślan et al., 2019*). Luminal $Ca^{2+}$ can bind to three non-canonical calcium-binding sites each formed by three acidic residues in AtTPC1 (site 1: D240, D454, and E528; site 2: E239, D240, and E457; site 3: E605, D606, and D607; *Dickinson et al., 2022*; *Guo et al., 2016*; *Kintzer and Stroud, 2016*). Site 1/2 residues are located in luminal linker regions between transmembrane domains while site 3 residues are found in the luminal pore entrance. Among them, sites 1 and 3 play key roles in voltage gating and luminal $Ca^{2+}$ inhibition.

Screening a chemically induced *Arabidopsis* mutant collection for individuals producing elevated amounts of the wound hormone jasmonate (JA) identified the TPC1 mutant *fou2* (fatty acid oxygenation upregulated 2) (*Bonaventure et al., 2007a*). Interestingly, the *fou2* channel behaves like a hyperactive TPC1 channel (*Beyhl et al., 2009*; *Bonaventure et al., 2007a*) and opens already close to the vacuole resting voltage which is around –30 mV (*Wang et al., 2015*). In contrast, wild-type AtTPC1 becomes active only upon depolarization. The hypersensitivity of the *fou2* TPC1 channel toward voltage confers hyperexcitability to the vacuole and results from a mutation of the negatively charged glutamate at position 454 to the uncharged asparagine (D454N) (*Bonaventure et al., 2007a*; *Jaślan et al., 2019*). This TPC1 site is directed toward the vacuole lumen and is part of the non-canonical $Ca^{2+}$ sensor site 1 (*Dadacz-Narloch et al., 2011*; *Guo et al., 2016*; *Kintzer and Stroud, 2016*). Therefore, the block of channel activity by elevated luminal $Ca^{2+}$ levels, typical for wild-type TPC1, is greatly attenuated in the *fou2* mutant (*Lenglet et al., 2017*). Consequently, *fou2* vacuoles must experience episodes in which the membrane potential is short circuited. As a result, *fou2* plants appear wounded and produce large amounts of jasmonate (*Bonaventure et al., 2007a*). Interestingly, strong membrane depolarizations triggered by current injection into healthy *Arabidopsis* leaves strongly activated the JA pathway, suggesting a link between vacuolar membrane potential and activation of JA synthesis (*Mousavi et al., 2013*). However, this link between TPC1-dependent vacuolar excitability and changes to electrical signaling of the plasma membrane and JA production remains to be explored (for review see *Pottosin and Dobrovinskaya, 2022*). Functional studies in the context of

the *fou2* mutation site showed that additional luminal amino acids other than the central $Ca^{2+}$-binding sites like D454 are involved in luminal $Ca^{2+}$ sensing, for example the adjacent E457 from $Ca^{2+}$ sensor site 2 (*Dadacz-Narloch et al., 2011*; *Guo et al., 2016*; *Kintzer and Stroud, 2016*). Neutralization of E457 desensitized AtTPC1 toward vacuolar $Ca^{2+}$ but unlike the *fou2* mutation did not additionally promote channel opening (*Dadacz-Narloch et al., 2011*).

The TPC1 channels from *Vicia faba* and *A. thaliana* are historically established prototypes of the SV channel. Nevertheless, only AtTPC1 but not VfTPC1 has been identified at the molecular level so far. To fil this gap, we cloned VfTPC1 and analyzed its electrical properties in the background of the AtTPC1-loss-of-function *Arabidopsis* mutant. Despite similarities of the fava bean channel with its *Arabidopsis* homolog, both TPC1s showed also astonishing differences in voltage- and luminal $Ca^{2+}$-dependent properties. In search for the structural reasons for this divergence, we could pinpoint three polymorphic sites within two luminal $Ca^{2+}$ coordinating regions. When these polymorphic sites were implemented into the *Arabidopsis* TPC1 channel, AtTPC1 was converted into a hyperactive SV/TPC1 channel desensitized to luminal $Ca^{2+}$, mimicking the properties of VfTPC1 and the AtTPC1 mutant *fou2*. Natural polymorphic TPC1 channel variants, like VfTPC1, with differences in key properties might be employed by some plant species to better meet their needs in specific environmental situations.

## Results
### VfTPC1 is a native TPC1 channel variant with a low luminal $Ca^{2+}$ sensitivity

In our search for natural TPC1 channel variants, we noticed that SV currents had been recorded from *V. faba* guard cell vacuoles even under unnatural high luminal $Ca^{2+}$ loads (50 mM; *Ward and Schroeder, 1994*). Considering that hardly any AtTPC1/SV currents were recorded in *A. thaliana* mesophyll vacuoles even at a lower luminal $Ca^{2+}$ level such as 10 mM (*Lenglet et al., 2017*), the TPC1 channel variants of *V. faba* and *A. thaliana* seem to differ in their $Ca^{2+}$ sensitivity. In contrast to *TPC1* from *A. thaliana* (*Peiter et al., 2005*), the *VfTPC1* gene has not yet been identified. To gain insights into the molecular structure and function of VfTPC1, we isolated RNA from fava bean. Following RNA-seq and de novo transcriptome assembly, we identified a single *VfTPC1* transcript, just as in *A. thaliana* (*Peiter et al., 2005*). After VfTPC1 was cloned and fused to an eYFP tag, mesophyll protoplasts isolated from the *Arabidopsis* TPC1-loss-of-function mutant *attpc1-2* were transiently transformed with the fava bean

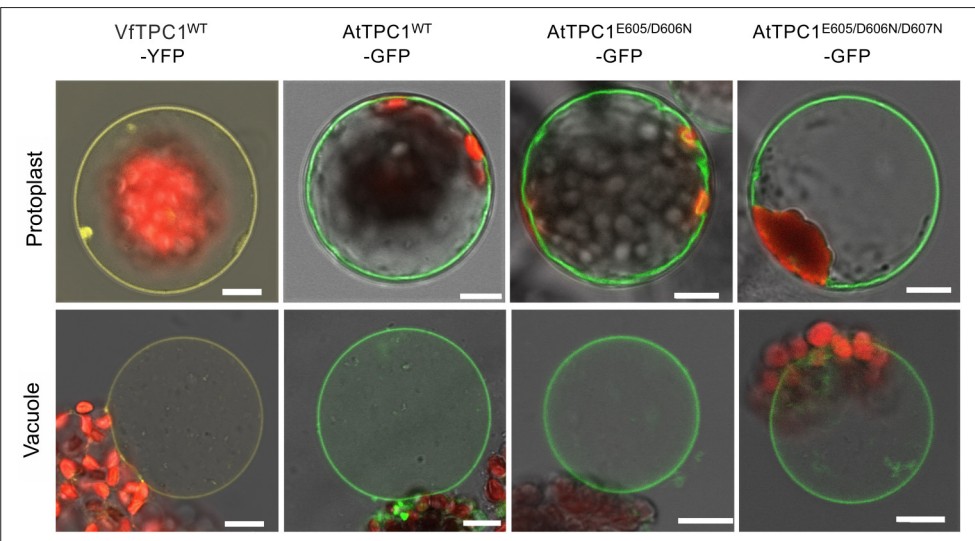

**Figure 1.** Vacuolar targeting of TPC1 channel variants. Mesophyll protoplasts from the *Arabidopsis thaliana* mutant *tpc1-2* after transient transformation with the respective GFP/YFP-tagged TPC1 channel construct and released vacuoles. Wild-type TPC1 constructs are indicated by WT. Bright field and fluorescent images were merged. Red fluorescence corresponds to chloroplast autofluorescence. The yellow and green fluorescence represent the YFP and GFP fluorescence, respectively. Scale bars = 10 μm.

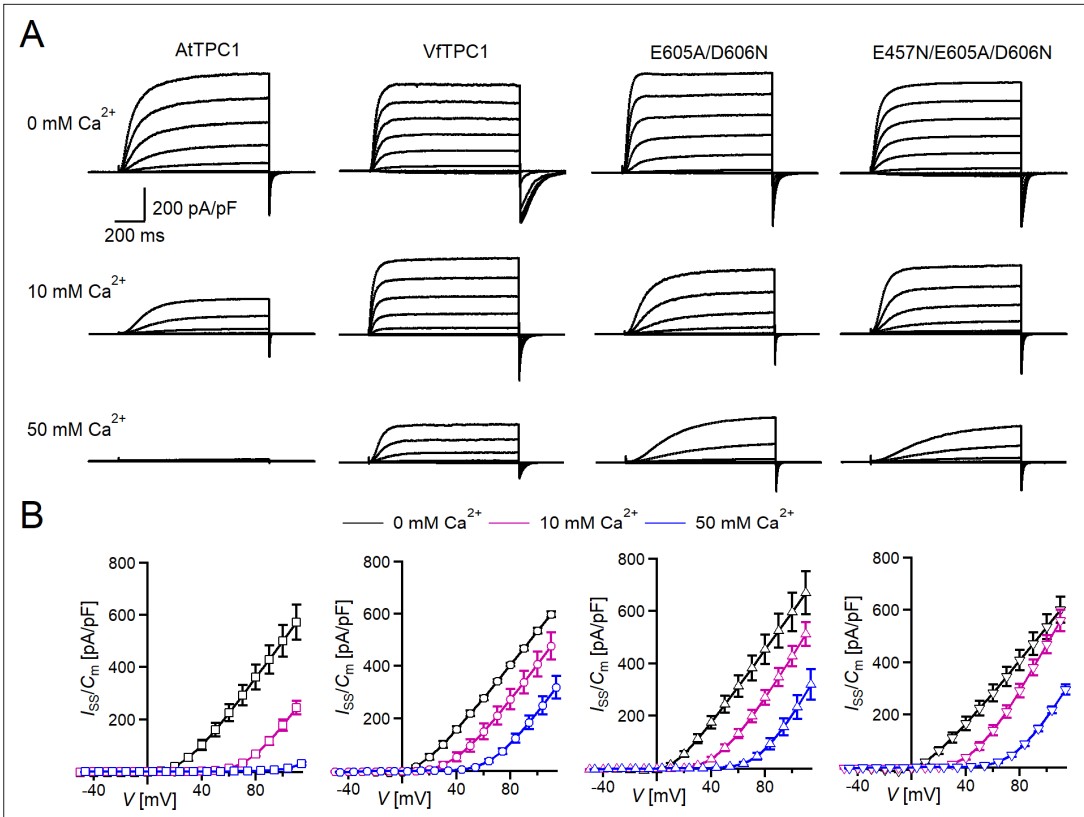

**Figure 2.** Effect of voltage and luminal $Ca^{2+}$ on TPC1/SV currents of *Vicia faba* and *Arabidopsis thaliana* channel variants. (**A**) Macroscopic TPC1/SV current recordings from mesophyll vacuoles liberated from *A. thaliana* protoplasts isolated from the TPC1-loss-of-function mutant *attpc1-2* and transformed with different TPC1 channel types. E605A/D606N and E457N/E605A/D606N represent AtTPC1 channel mutants. AtTPC1 and VfTPC1 denote wild-type TPC1 channels from *A. thaliana* and *V. faba*, respectively. TPC1/SV currents elicited upon depolarizing voltages pulses in the range −80 to +110 mV in 20 mV increments at indicated luminal $Ca^{2+}$ concentrations are shown. Before and after these voltage pulses, the membrane was clamped to the holding voltage of −60 mV. (**B**) Normalized TPC1/SV steady-state currents ($I_{SS}/C_m$) derived from current recordings under different luminal $Ca^{2+}$ conditions as those shown in (**A**) were plotted against the clamped membrane voltage (*V*). Symbols represent means ± SE. Squares = AtTPC1 wild type with $n_{0/10Ca} = 5$, $n_{50Ca} = 4$; circles = VfTPC1 wild type with $n_{0Ca} = 5$, $n_{10/50Ca} = 6$; upright triangles = AtTPC1-E605A/D606N with $n_{0/10Ca} = 5$, $n_{50Ca} = 4$; reversed triangles = AtTPC1-E457N/E605A/D606N with $n_{0/10Ca} = 5$, $n_{50Ca} = 4$. In (**B**), AtTPC1 wild-type data at 0 and 10 mM $Ca^{2+}$ are identical to those shown in *Dickinson et al., 2022*; Creative Commons Attribution-Non Commercial-NoDerivatives License 4.0; CC BY-NC-ND. Experiments in (**A, B**) were performed under symmetric $K^+$ conditions (150 mM) with 1 mM cytosolic $Ca^{2+}$ and luminal $Ca^{2+}$ at the indicated concentration. For further details on the voltage pulse protocol and solutions, see Materials and methods.

The online version of this article includes the following source data and figure supplement(s) for figure 2:

**Source data 1.** Quantification of normalized steady-state current amplitudes of *Arabidopsis thaliana* TPC1 channel variants and *Vicia faba* TPC1 expressed in the *Arabidopsis* mutant *attpc1-2*.

**Figure supplement 1.** Species-dependent effect on voltage activation threshold of TPC1/SV currents.

---

TPC1 channel. Similar to TPC1 from *Arabidopsis* (*Figure 1*) and other plant species (*Dadacz-Narloch et al., 2013*), VfTPC1 was found to localize exclusively to the vacuole membrane as visualized by the fluorescent eYFP signal (*Figure 1*). In whole-vacuole patch-clamp experiments with such fluorescent mesophyll vacuoles, macroscopic outward-rectifying SV/TPC1-like currents were elicited upon depolarizing voltage pulses (*Figure 2*). However, compared to AtTPC1, VfTPC1 channels differed in kinetics, voltage dependence, and luminal calcium sensitivity (*Figures 2–5*).

In the presence of physiological-like luminal $Mg^{2+}$ (2 mM) (*Shaul, 2002*) but absence of luminal $Ca^{2+}$, the current activation was faster and the current deactivation slower with VfTPC1 than with

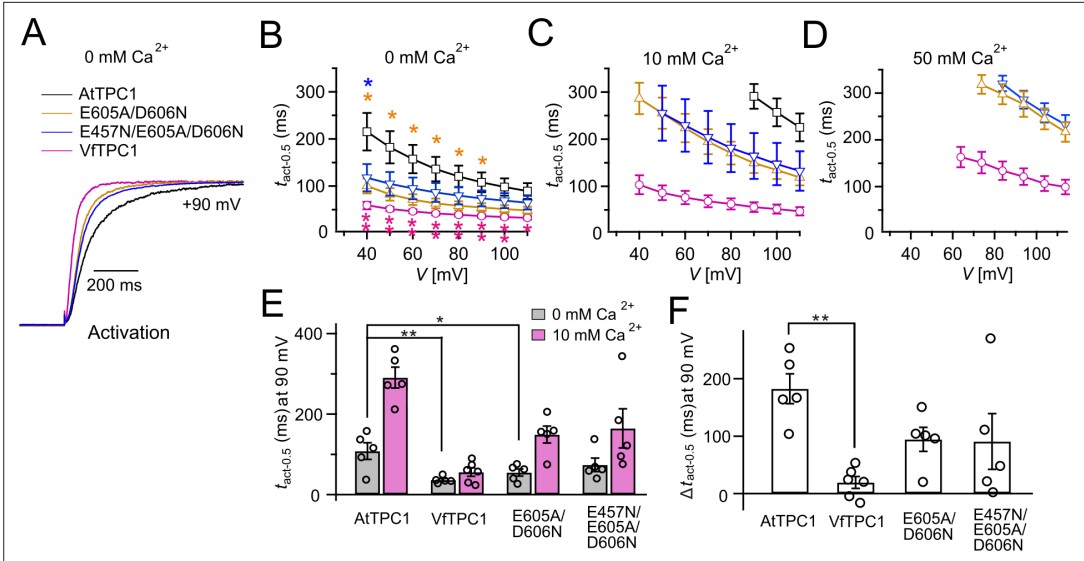

**Figure 3.** Half-activation times of TPC1 channel variants. (**A**) Representative current relaxation induced upon a voltage pulse from the holding voltage of −60 to +90 mV at 0 mM luminal $Ca^{2+}$. Normalized current responses of the vacuoles equipped with one of the indicated TPC1 channel variants were superimposed. (**B–D**) Half-activation times ($t_{act-0.5}$) of the voltage-induced TPC1 currents at indicated luminal $Ca^{2+}$ concentrations. In (**B**), stars denote that the $t_{act-0.5}$ values of VfTPC1 (magenta), the ATPC1 double mutant E605A/D606N (yellow), and AtTPC1 triple mutant (blue) significantly differ from AtTPC1 wild type (one-way analysis of variance [ANOVA] together with a Dunnett's post hoc comparison test; *$p < 0.05$; **$p < 0.01$). (**E**) Comparison of half-activation times determined at +90 mV for 0 and 10 mM luminal $Ca^{2+}$. $t_{act-0.5}$ values under 0 mM luminal $Ca^{2+}$ were tested for significant differences with one-way ANOVA combined with a Dunnett's post hoc comparison test (*$p < 0.05$; **$p < 0.01$). (**F**) The differences in the half-activation times from (**E**) between 0 and 10 luminal $Ca^{2+}$ (one-way ANOVA together with a Dunnett's post hoc comparison test; **$p < 0.01$). In (**B–F**) the number of experiments (*n*) was identical with *Figure 2B*, and data are represented as means ± SE. In (**E, F**), individual data points were additionally inserted as open black circles into the bar chart. Experiments in (**A–F**) were performed with vacuoles from *attpc1-2* mesophyll protoplasts transiently transformed with the indicated TPC1 channel variants. Symmetric $K^+$ conditions (150 mM) were used with 1 mM cytosolic $Ca^{2+}$ and luminal $Ca^{2+}$ at indicated concentration. For details on the voltage pulse protocol and solutions, see Materials and methods.

The online version of this article includes the following source data for figure 3:

**Source data 1.** Quantification of current activation kinetics of *Arabidopsis thaliana* TPC1 channel variants and *Vicia faba* TPC1 expressed in the *Arabidopsis* mutant *attpc1-2*.

AtTPC1 (*Figures 2A, 3A, B, 4A, B*). A closer look at the current–voltage curves further revealed that in addition to outward currents, inward currents were also triggered by voltages in the range of −40 and 0 mV, but only from vacuoles harboring VfTPC1 and not AtTPC1 (*Figure 2—figure supplement 1*). This points to a shift in the voltage activation threshold for VfTPC1 channel opening by about 30 mV to more negative voltages. For further quantification of this effect, the voltage-dependent relative open-channel probability curves ($G/G_{max}(V)$) were determined for both TPC1 channel variants (*Figure 5A*). They were fitted with a double Boltzmann equation describing the voltage-dependent channel transitions between two closed and one open state ($C_2 \leftrightarrows C_1 \leftrightarrows O$) (*Jaślan et al., 2016*; *Pottosin et al., 2004*). From these fits the midpoint voltages $V_2$ and $V_1$ were derived (*Figure 5B*), showing in particular, a significant difference in midpoint voltage $V_1$ of VfTPC1 (−3.5 ± 12.3 mV, *n* = 5) and AtTPC1 (51.2 ± 9.6 mV, *n* = 5). These results confirm that VfTPC1, unlike AtTPC1, activates near the vacuolar resting membrane voltage (*Figure 5B*).

When the luminal $Ca^{2+}$ concentration was increased from 0 to 10 mM $Ca^{2+}$, the activation kinetics of AtTPC1 was strongly slowed down (*Figure 3B, C, E, F*). However, this effect was much less pronounced for VfTPC1. At +90 mV, for example, the half-activation time increased by about 170% for AtTPC1, but only 53% for VfTPC1 (*Figure 3A–C, E, F*). Even more impressive was the fact that AtTPC1 currents were suppressed by about 50% at 10 mM luminal $Ca^{2+}$ and completely vanished at 50 mM luminal $Ca^{2+}$ (*Figure 2*), the concentration used in the very early work on SV currents from

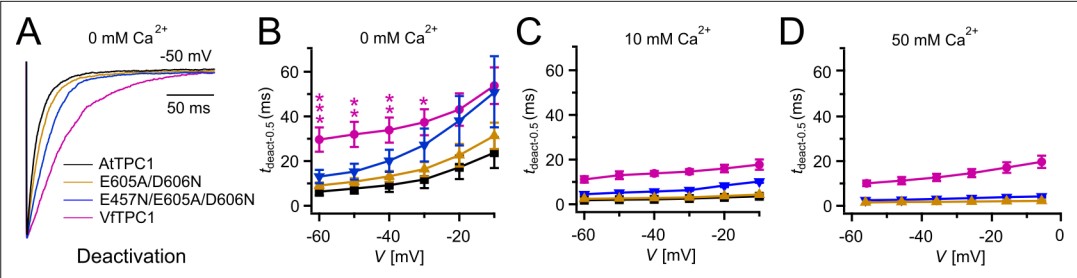

**Figure 4.** Half-deactivation times of TPC1 channel variants. (**A**) Representative current relaxation induced upon a voltage pulse from +80 mV (VfTPC1) or +100 mV (AtTPC1 channel variants) to −50 mV at 0 mM luminal $Ca^{2+}$. Normalized current responses of the vacuoles equipped with one of the indicated TPC1 channel variants were superimposed. (**B–D**) Half-deactivation times ($t_{deact-0.5}$) of the TPC1-mediated slow vacuolar (SV) currents at indicated luminal $Ca^{2+}$ concentrations plotted against the respective membrane voltages. In (**B**), stars denote significant differences between VfTPC1 and AtTPC1 wild type (one-way analysis of variance [ANOVA] together with a Dunnett's post hoc comparison test; *$p < 0.05$; **$p < 0.01$; ***$p < 0.001$). In (**B–D**), data represent means ± SE, and the number of experiments ($n$) was as follows: AtTPC1 wild type $n_{0Ca} = 5$, $n_{10Ca} = 3$; VfTPC1 wild type $n_{0/10Ca} = 4$, $n_{50Ca} = 6$; AtTPC1-E605A/D606N $n_{0/50Ca} = 4$, $n_{10Ca} = 5$; AtTPC1-E457N/E605A/D606N $n_{0/10Ca} = 5$, $n_{50Ca} = 4$. Experiments in (**A–D**) were performed with vacuoles from *attpc1-2* mesophyll protoplasts transiently transformed with the indicated TPC1 channel variants. Symmetric $K^+$ conditions (150 mM) were used with 1 mM cytosolic $Ca^{2+}$ and luminal $Ca^{2+}$ at the indicated concentration. For details on the double-voltage pulse protocol and solutions, see Materials and methods.

The online version of this article includes the following source data for figure 4:

**Source data 1.** Quantification of current deactivation kinetics of *Arabidopsis thaliana* TPC1 channel variants and *Vicia faba* TPC1 expressed in the *Arabidopsis* mutant *attpc1-2*.

fava bean (*Ward and Schroeder, 1994*). In contrast, the VfTPC1 current densities at zero and 10 mM luminal $Ca^{2+}$ were similar in magnitude, and at 50 mM $Ca^{2+}$ the VfTPC1 current density still reached about 50% of the level measured under luminal $Ca^{2+}$-free conditions. The strongly reduced susceptibility of VfTPC1 currents to inhibitory luminal $Ca^{2+}$ ions was associated with a significantly reduced inhibitory effect on voltage activation (*Figure 5*). Compared to 0 mM $Ca^{2+}$, at 10 mM $Ca^{2+}$ the $V_1$ and $V_2$ values increased by about 115 and 61 mV for AtTPC1, but only by about 61 and 26 mV for VfTPC1, respectively (*Figure 5B, C*). A rise from 10 to 50 mM luminal $Ca^{2+}$ caused a further positive-going shift of the VfTPC1 activation voltage curve ($G/G_{max}(V)$), indicated by a 1.7- and 1.6-fold rise in $V_1$ and $V_2$, respectively (*Figure 5B*). AtTPC1 currents, however, were so small at 50 mM luminal $Ca^{2+}$ (*Figure 2*) that activation voltage curves ($G/G_{max}(V)$) could not be resolved reliably. Together these findings document that VfTPC1 is much less susceptible to inhibitory luminal $Ca^{2+}$ than AtTPC1.

## Fabaceae and Brassicaceae TPC1 channels are polymorphic in luminal $Ca^{2+}$-sensing motifs

To gain insights into the functional domains of these Brassicaceae and Fabaceae TPC1 channel proteins underlying their different response to membrane voltage and luminal $Ca^{2+}$, we aligned not only the AtTPC1 and VfTPC1 but also other TPC1-like amino acid sequences of these plant families. Focusing on charged residues at the three sites that are involved in luminal $Ca^{2+}$ coordination and sensitivity in AtTPC1 (*Figure 5*; *Dadacz-Narloch et al., 2011*; *Dickinson et al., 2022*; *Kintzer and Stroud, 2016*), we found that these sites were highly conserved among the Brassicaceae TPC1s. Only two of the 12 TPC1 variants tested (BrTPC1 and LeTPC1) exhibit a neutralizing substitution at sites homologous to residue D606 at the luminal pore entrance of AtTPC1. In contrast, most TPC1 types of the Fabaceae species show one up to three non-conservative variations of the residues. Eleven of the 14 Fabaceae TPC1 channels harbor a non-charged asparagine or glutamine instead of the negatively charged glutamate at sites homologous to AtTPC1-E456 or -E457 (*Figure 6*). Remarkably, this E456Q/E457N polymorphism was additionally accompanied by the neutralization of one or two negatively charged residues at site 3 within the luminal pore entrance in all Fabaceae TPC1 channels, with the exception of *Bauhinia tomentosa* BtTPC1 (*Figure 6*). In this regard, the Fabaceae TPC1 channels group in two clusters, containing either one or two non-charged residues (Ala/Val, Asn) at the homologous sites to

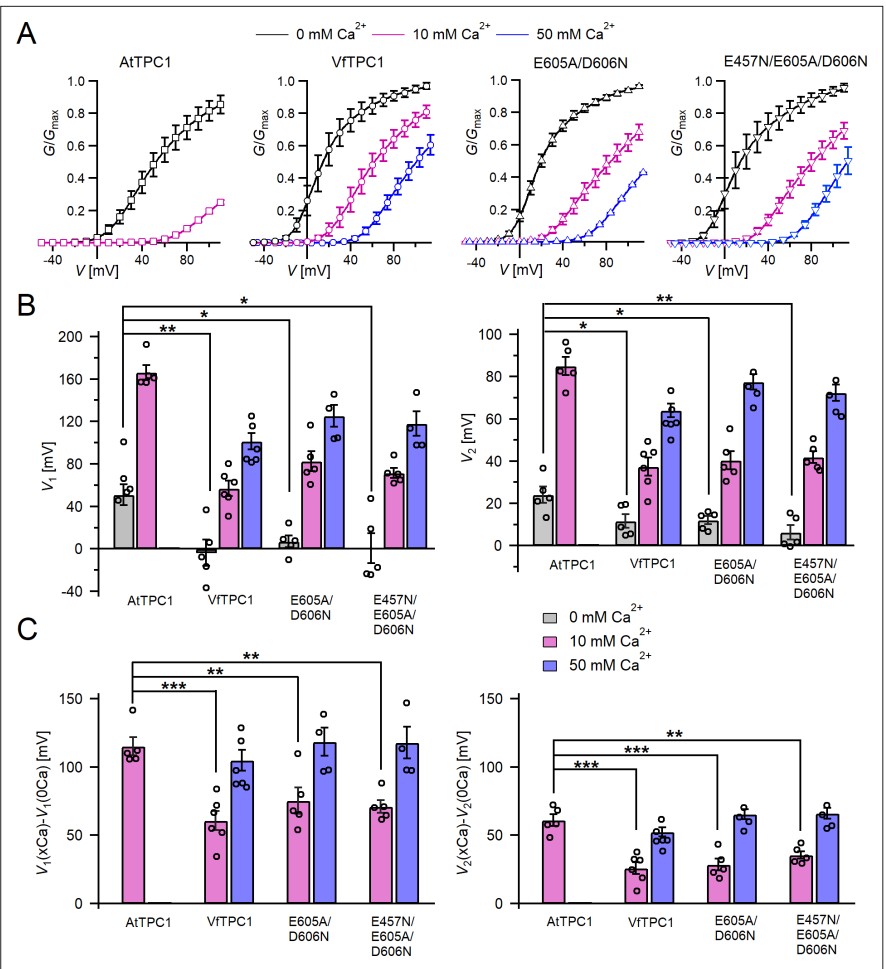

**Figure 5.** Channel activity of *Vicia faba* and *Arabidopsis thaliana* TPC1 channel variants in response to membrane voltage and luminal $Ca^{2+}$. (**A**) Conductance–voltage plots ($G/G_{max}(V)$) determined for the different TPC1 channel variants as a measure for their relative open-channel probability under indicated luminal $Ca^{2+}$ conditions. Best fits of the $G/V$ plots to a double Boltzmann function are given by the solid lines. Squares = AtTPC1 wild type, circles = VfTPC1 wild type, upright triangles = AtTPC1-E605A/D606N, reversed triangles = AtTPC1-E457N/E605A/D606N. (**B**) The midpoint voltages $V_1$ (left) and $V_2$ (right) derived from the fits of the $G/V$ plots shown in (A) are given for the different channel variants at the indicated $Ca^{2+}$ condition. To test for significant differences between the $V_{1/2}$ values under 0 mM luminal $Ca^{2+}$, a statistical analysis was performed with one-way analysis of variance (ANOVA) combined with a Dunnett's post hoc comparison test (*p <0 .05, **p < 0.01). (**C**) The differences in the midpoint voltages $V_1$ (left) and $V_2$ (right) shown in (**B**) between 0 and 10 and if available between 0 and 50 luminal $Ca^{2+}$ are shown. The changes in $V_{1/2}$ values related to a rise from 0 to at 10 $Ca^{2+}$ were statistically analyzed with one-way ANOVA together with a Dunnett's post hoc comparison test (**p < 0.01; ***p <0 .001). In (**B**) and (**C**), individual data points were inserted as open black circles into the bar chart. In (**A–C**), the number of experiments (*n*) was as follows: AtTPC1 wild type $n_{0/10Ca}$ = 5; VfTPC1 wild type with $n_{0Ca}$ = 5, $n_{10/50Ca}$ = 6; AtTPC1-E605A/D606N $n_{0/10Ca}$ = 5, $n_{50Ca}$ = 4; AtTPC1-E457N/E605A/D606N $n_{0/10Ca}$ = 5, $n_{50Ca}$ = 4. Data in (**A–C**) represent means ± SE. AtTPC1 wild-type data at 0 and 10 mM $Ca^{2+}$ are identical to those shown in *Dickinson et al., 2022* (Creative Commons Attribution-NonCommercial-NoDerivatives License 4.0; CC BY-NC-ND). Experiments in (**A–C**) were performed with vacuoles from *attpc1-2* mesophyll protoplasts transiently transformed with the indicated TPC1 channel variants. Symmetric $K^+$ conditions (150 mM) were used with 1 mM cytosolic $Ca^{2+}$ and luminal $Ca^{2+}$ at the indicated concentration. For details on the analysis of tail current recordings, the associated pulse protocol and solutions, see Materials and methods.

The online version of this article includes the following source data for figure 5:

**Source data 1.** Quantification of normalized conductance/voltage curves of *Arabidopsis thaliana* TPC1 channel variants and *Vicia faba* TPC1 expressed in the *Arabidopsis* mutant *attpc1-2*.

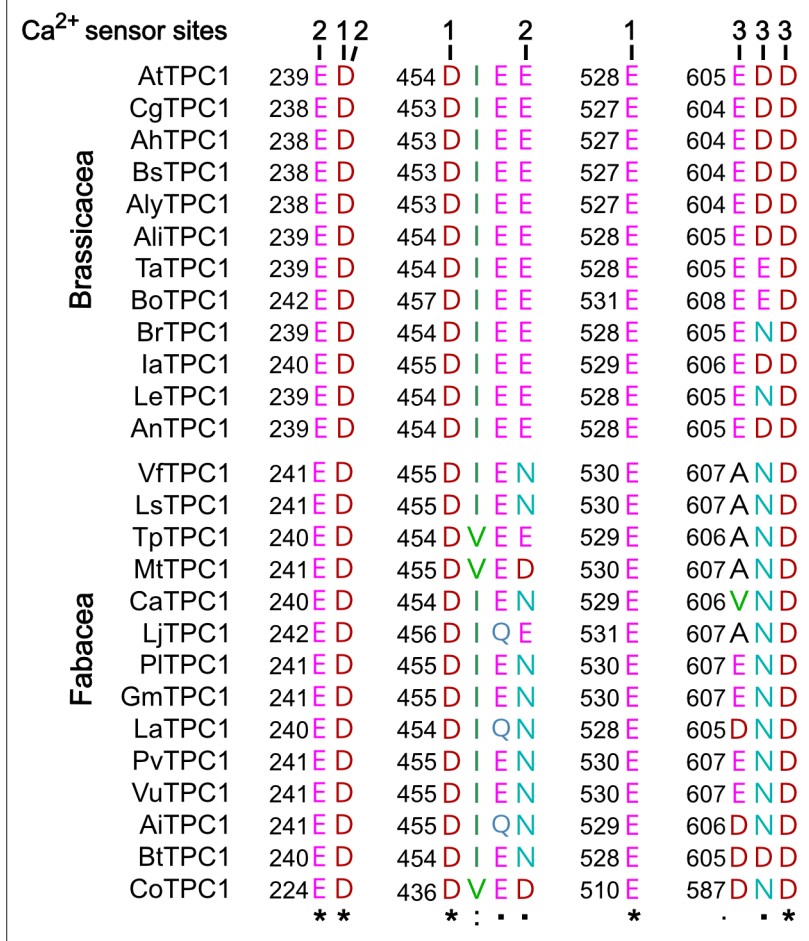

**Figure 6.** Polymorphism of functional TPC1 channel domains with a role in luminal $Ca^{2+}$ coordination. Sections of an amino acid sequence alignment of Brassicacea and Fabacea TPC1 channels in the region of the three $Ca^{2+}$ sensor sites. The numbers above the residues of AtTPC1 indicate to which $Ca^{2+}$ sensor site they contribute. A different color code was used for different amino acids. An asterisk marks 100% conserved residues across the sequences while a colon (:) indicates a conservative and a dot (.) denotes a non-conservative substitution. TPC1 sequences from following species were used: *Arabidopsis thaliana* (AtTPC1), *Capsella grandiflora* (CgTPC1), *Arabidopsis halleri* (AhTPC1), *Boechera stricta* (BsTPC1), *Arabidopsis lyrata* (AlyTPC1), *Alyssum linifolium* (AliTPC1), *Thlaspi arvense* (TaTPC1), *Brassica oleracea* (BoTPC1), *Brassica rapa* (BrTPC1), *Iberis amara* (IaTPC1), *Lepidium sativum* (LeTPC1), *Arabis nemorensis* (AnTPC1), *Vicia faba* (VfTPC1), *Lathyrus sativus* (LsTPC1), *Trifolium pratense* (TpTPC1), *Medicago truncatula* (MtTPC1), *Cicer arietinum* (CaTPC1), *Lotus japonicus* (LjTPC1), *Phaseolus lunatus* (PlTPC1), *Glycine max* (GmTPC1), *Lupinus albus* (LaTPC1), *Phaseolus vulgaris* (PvTPC1), *Vigna unguiculata* (VuTPC1), *Arachis ipaensis* (AiTPC1), *Bauhinia tomentosa* (BtTPC1), and *Copaifera officianalis* (CoTPC1). Amino acid sequences and their sources are listed in *Figure 6—source data 1*.

The online version of this article includes the following source data and figure supplement(s) for figure 6:

**Source data 1.** Amino acid sequences of plant TPC1 channels and their accession numbers.

**Figure supplement 1.** Consensus tree of Brassicacea and Fabacea TPC1 channel proteins.

---

AtTPC1-E605 and -D606 (*Figure 6—figure supplement 1*). The fava bean TPC1 channel belongs to a triple-polymorphism subgroup, containing alanine and asparagine at sites homologous to the $Ca^{2+}$ sensor sites 2 and 3 of AtTPC1 (*Figure 6*; AtTPC1-E457/E605/D606; VfTPC1-N458/A607/N608).

## The polymorphic AtTPC1 double mutant E605A/D606N mimics the VfTPC1 channel features

To clarify whether these three polymorphic sites are responsible for the different gating behavior of the two TPC1 channel variants, a series of site-directed mutagenesis was initiated in both VfTPC1

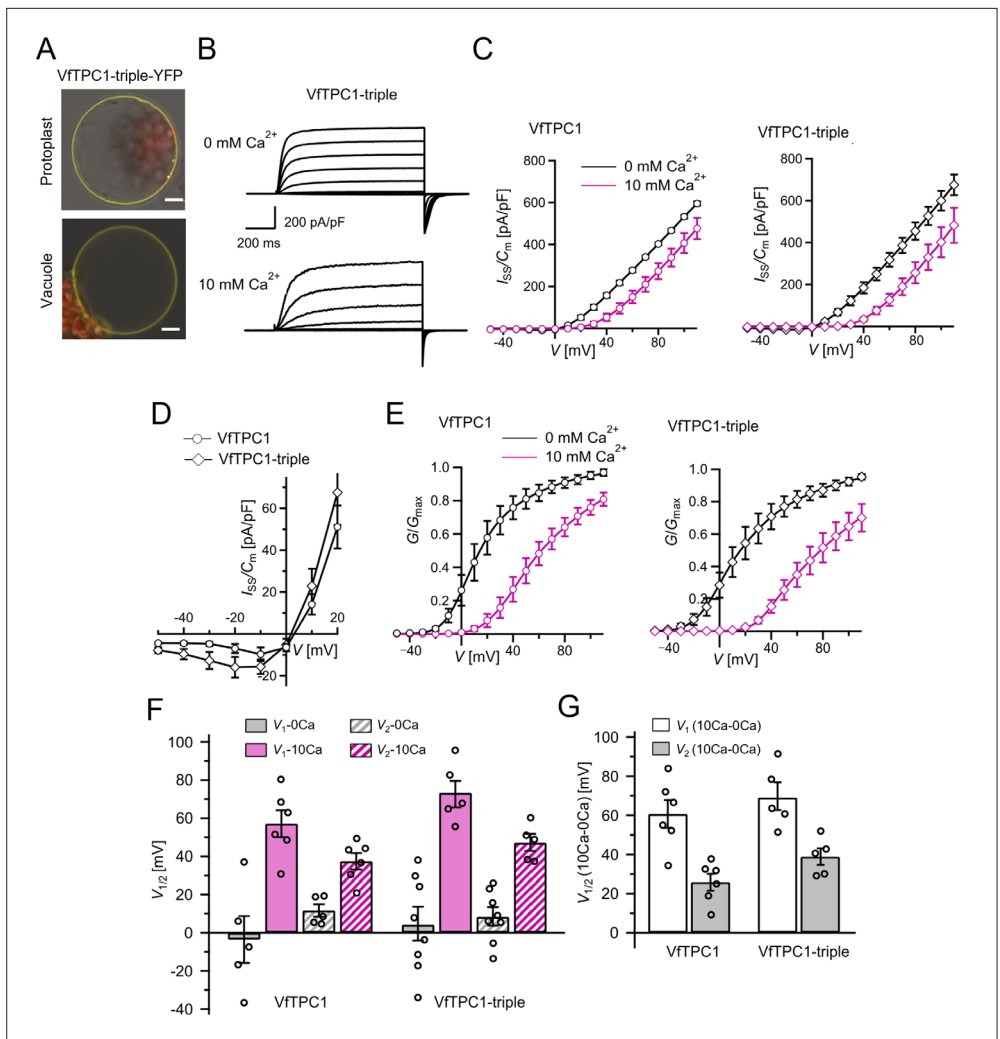

**Figure 7.** The voltage and luminal $Ca^{2+}$ dependence of the VfTPC1 triple mutant N458E/A607E/N608D. (**A**) Mesophyll vacuoles released from *attpc1-2* mesophyll protoplasts after transient transformation with eYFP-tagged VfTPC1 triple mutant construct. Bright field and fluorescent images were merged. Red and yellow fluorescence corresponds to chloroplast autofluorescence and eYFP fluorescence, respectively. Scale bars = 10 μm. (**B**) Macroscopic TPC1 current recordings from *attpc1-2* mesophyll vacuoles transformed with the VfTPC1 triple mutant. (**C**) Normalized steady-state currents ($I_{ss}/C_m(V)$) determined for the indicated VfTPC1 channel variant and plotted against the clamped membrane voltages in the presence (10 mM) and absence of luminal $Ca^{2+}$. (**D**) Enlarged section of the current–voltage curves ($I_{ss}/C_m(V)$) for VfTPC1 channel variants shown in (**C**) at 0 mM luminal $Ca^{2+}$. (**E**) Normalized conductance–voltage curves ($G/G_{max}(V)$) of VfTPC1 channel variants in the presence and absence of luminal $Ca^{2+}$. (**F**) Midpoint activation voltages $V_{1/2}$ are given for the different channel variants at the indicated $Ca^{2+}$ condition. VfTPC1 triple mutant with $n_{0Ca} = 8$ and $n_{10Ca} = 5$; VfTPC1 wild type with $n_{0Ca} = 5$, $n_{10Ca} = 6$. The data of VfTPC1 wild type in (**C–G**) are identical to those in *Figures 2 and 5* and *Figure 2—figure supplement 1* and were shown only for better direct comparison. Data points in (**C–G**) represent means ± SE. In (**F**) and (**G**), individual data points were inserted as open black circles into the bar chart. According to the statistical analysis (t-test), the $V_1/V_2$ values at 0 mM $Ca^{2+}$ (**F**) and the $Ca^{2+}$-induced shift in $V_1/V_2$ values (**G**) between VfTPC1 wild-type and the VfTPC1 triple mutant were not significantly different, as the p-values were > 0.05. For details of the statistical analysis, see . Source data for patch-clamp data of the VfTPC1 triple mutant are provided in while those of VfTPC1 wild type are found in *Figure 2—source data 1* and *Figure 5—source data 1*.

The online version of this article includes the following source data for figure 7:

**Source data 1.** Quantification of normalized steady-state current/voltage curves and normalized conductance/voltage curves of Vicia faba TPC1 triple mutant expressed in the Arabidopsis mutant attpc1-2.

and AtTPC1. This resulted in the triple VfTPC1 mutant N458E/A607E/N608D and the double, triple AtTPC1 mutants E605A/D606N and E457N/E605A/D606N, respectively. When transiently expressed in the background of the *attpc1-2* mutant (*Figure 1* and *Figure 7A*), all these channel mutants were localized in the vacuole membrane and gave rise to the typical SV/TPC1 channel currents (*Figures 2A and 7B*). When the two VfTPC1 channel variants were examined with respect to their voltage dependence and luminal $Ca^{2+}$ sensitivity, no significant differences were found between VfTPC1 wild type and its triple mutant (*Figure 7C–G*). In contrast, the AtTPC1 mutants differed in some respects from AtTPC1 wild type. Compared to AtTPC1 wild type, the double but not the triple AtTPC1 mutant activated significantly faster under luminal $Ca^{2+}$-free conditions (*Figure 3*). The lack of effect on activation of the AtTPC1 triple mutant is likely related to the attenuating influence of the mutated residue E457N, because *Dadacz-Narloch et al., 2011* showed that the activation kinetics of the single AtTPC1 mutant E457N were slowed down. The deactivation of both, the double and triple AtTPC1 mutants, however, was not much affected and was more AtTPC1- than VfTPC1-like (*Figure 4*), suggesting that other functional domains than $Ca^{2+}$ sensor site 3 determines the deactivation behavior of TPC1 channels. This view is also supported by the fact that deactivation of the PpTPC1a channel from the moss *Physcomitrella patens* was slower than that of AtTPC1, although all three luminal $Ca^{2+}$ sensor sites 1–3 were functional (*Figure 8*) and resulted in AtTPC1-like luminal $Ca^{2+}$ sensitivity and voltage dependence of PpTPC1a (*Dadacz-Narloch et al., 2011*). In addition, in the absence of luminal $Ca^{2+}$, the current voltage curves ($I_{ss}/C_m(V)$) and even more so the activation voltage curves ($G/G_{max}(V)$) of the double and triple AtTPC1 mutants appeared shifted to more negative voltages by about 30 mV compared to wild-type AtTPC1 (*Figures 2B and 5A*, *Figure 2—figure supplement 1*). The changed voltage dependence of both AtTPC1 mutants was reflected in $V_1$ and $V_2$ values similar to VfTPC1 (*Figure 5B*). When the two AtTPC1 mutants faced 10 mM luminal $Ca^{2+}$, again the current and activation voltage curves as well as the $V_1$, $V_2$ values resembled those of VfTPC1 (*Figures 2B and 5*). The fact that AtTPC1 channels carrying the VfTPC1 polymorphisms (i.e., double or triple residue substitutions) exhibit a low luminal $Ca^{2+}$ susceptibility is best displayed by the channel behavior to 50 mM $Ca^{2+}$. Under such high luminal $Ca^{2+}$ loads, almost no SV/TPC1 currents were observed with wild-type AtTPC1 (*Figure 2*). The current and voltage activation curves ($I_{ss}/C_m(V)$, $G/G_{max}(V)$) of the AtTPC1 mutants and VfTPC1, however, showed similar strong SV channel activity even at this extreme luminal $Ca^{2+}$ level (*Figures 2B and 5*). Thus, the E605A/D606N exchange is already sufficient to provide AtTPC1 with the voltage and $Ca^{2+}$ sensitivity of VfTPC1. The additional E457N replacement in the AtTPC1 E605A/D606N mutant background, however, did not further affect the transition from an *Arabidopsis*-like to a VfTPC1-like gating behavior, nor did it further reduce luminal $Ca^{2+}$ sensitivity. *Dadacz-Narloch et al., 2011* reported that neutral residues at site 457 (E → N/Q) in AtTPC1 made the channel significantly less sensitive to luminal $Ca^{2+}$ but attenuated voltage-dependent activation compared to wild-type channels, as midpoint voltages ($V_1$, $V_2$) were shifted to more positive values. Thus, the hyperactivity of the native *V. faba* TPC1 channel is most likely related to the altered $Ca^{2+}$ sensor site 3 rather than to $Ca^{2+}$ sensor site 2. Considering further the similar luminal $Ca^{2+}$ sensitivity

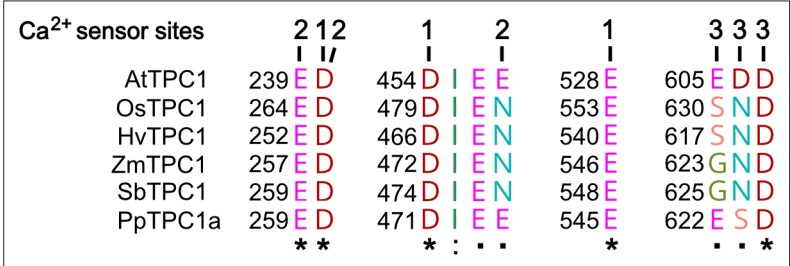

**Figure 8.** Amino acid sequence comparison of TPC1 channels from different species in the region of the three $Ca^{2+}$ sensor sites. The numbers above the residues of AtTPC1 indicate to which $Ca^{2+}$ sensor site they contribute. A different color code was used for different amino acids. An asterisk marks 100% conserved residues across the sequences while a colon (:) indicates a conservative and a dot (.) denotes a non-conservative substitution. TPC1 sequences from following species were used: *Arabidopsis thaliana* (AtTPC1), *Oryza sativa* (OsTPC1), *Hordeum vulgaris* (HvTPC1), *Zea mays* (ZmTPC1), *Sorghum bicolor* (SbTPC1), and *Physcomitrella patens* (PpTPC1a). Amino acid sequences and their sources are listed in *Figure 6—source data 1*.

of E605A/D606N and E457N/E605A/D606N (*Figure 5C*), the two Ca²⁺ sensor sites 2 and 3 with their residues E457 and E605/D606, respectively, do not have additive effects on channel gating.

## 3D topology of VfTPC1 and AtTPC1 triple mutant mimics that of *fou2*

In an attempt to determine the structural basis for the functional differences between *V. faba* and *A. thaliana* TPC1, we aimed to determine the cryoEM structure of VfTPC1, expressed and purified by similar conditions to that of AtTPC1 (*Dickinson et al., 2022*). Surprisingly, the biochemical behavior of VfTPC1 was significantly different than that of AtTPC1 and we were unable to recover any usable material after purification. Hence, we constructed a homology model of VfTPC1 using our high-resolution cryoEM structure of Ca²⁺-bound AtTPC1-D454N (*fou2*) as a template (*Figure 9*, *Supplementary file 1*). From our previous structures of wild-type AtTPC1 and *fou2* (*Dickinson et al., 2022*; *Kintzer and Stroud, 2016*), we determined that the luminal pore entrance operates an inhibitory

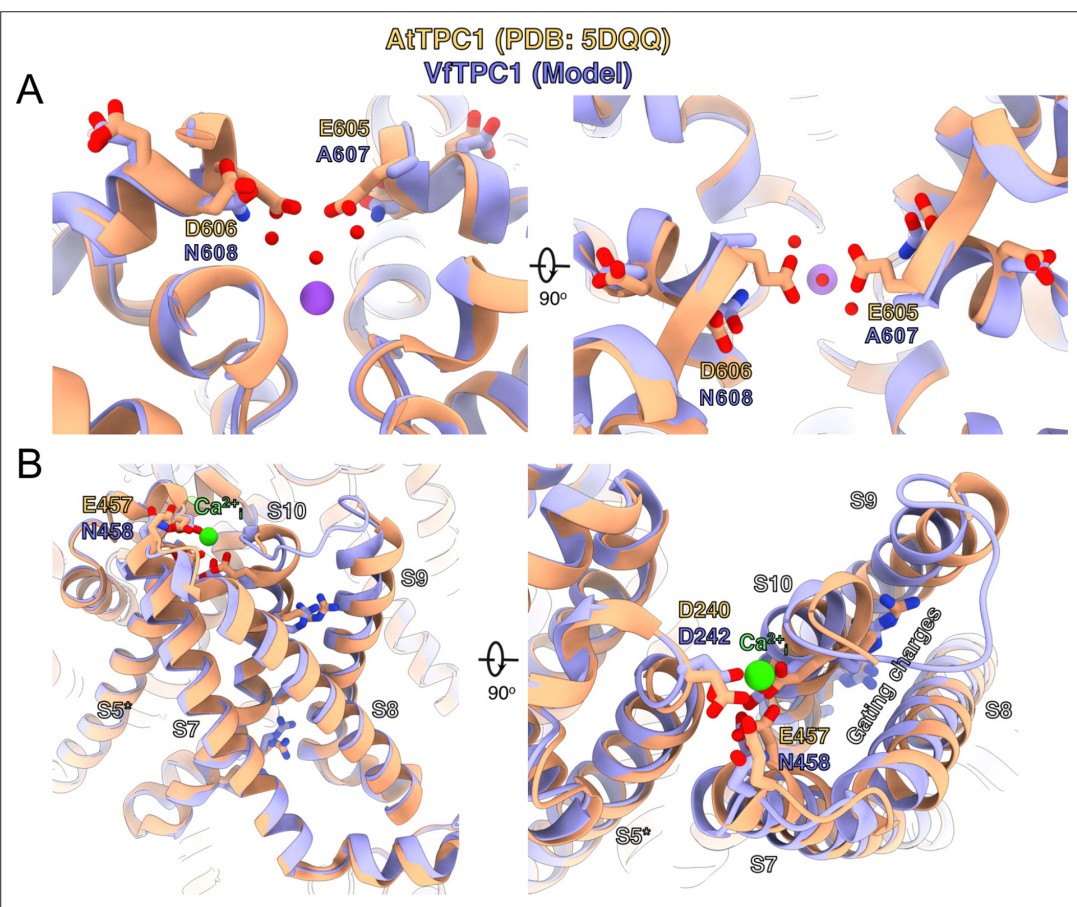

**Figure 9.** Structural comparison between wild-type AtTPC1 structure (beige color) and a VfTPC1 homology model (purple) depicts the functional consequences of sequence divergence. (**A**) Left: side view from in the plane of the membrane of the Ca²⁺ site showing the external (luminal) pore entry at the top. Right: top view from the external (luminal) surface of the Ca²⁺ site (90° from the left-side image) showing positions of key pore mouth residues. The central purple spheres represent Na⁺ ions. Red spheres are waters, as observed in the cryoEM structure of *fou2* (*Dickinson et al., 2022*). (**B**) Coordination of the luminal Ca²⁺-binding site (green sphere) on voltage-sensing domain (VSD2). Left: side view from the plane of the membrane of the voltage sensor domain, looking in toward the vertical axis of the pore. S8 helix is closest to the viewing direction. Right: top view from the external surface of the voltage sensor domain. The image is rotated 90° to obtain the top view from the left panel, and rotated in plane of the page such that S8 is on the right side. The gating charges on helix S10 are indicated by labeling. The model is provided by *Supplementary file 1*. An overview of the structure seen from the plane of the membrane is shown in *Figure 9—figure supplement 1*.

The online version of this article includes the following figure supplement(s) for figure 9:

**Figure supplement 1.** Topology and structural model of VfTPC1.

$Ca^{2+}$ sensor (site 3) that is coupled to the functional VSD2. In the wild-type and *fou2* structures, the three luminal acidic residues (E605, D606, and D607) of site 3 line the conduction pathway upstream of the selectivity filter (*Figure 9—figure supplement 1*). In the wild-type crystal structure, E605 binds a divalent metal on the symmetry axis. In our high-resolution structure of *fou2* (*Dickinson et al., 2022*), we showed that these residues rearrange to interact with the hydration shell around a bound metal in the selectivity filter and that D606 in *fou2* moves to the position of E605 in the wild-type structure, suggesting that both acidic residues regulate channel function. In our homology model of VfTPC1, neutralization of the acidic residues A607 and N608 in the luminal pore entrance (equivalent to E605 and D606 in AtTPC1) clearly alters the electrostatics of the pore. These substitutions almost certainly prevent the pore entrance from binding the inhibitory $Ca^{2+}$ observed in the wild-type structure (*Figure 9A*, *Figure 9—figure supplement 1*), and from interacting with the water network in the pore as seen in *fou2* (*Dickinson et al., 2022*). Recent MD simulations with AtTPC1 (*Navarro-Retamal et al., 2021*) additionally indicated that the E605/D606 motif is structurally very flexible and adapts according to the direction of ion flow. This may suggest a valve-like regulatory mechanism: The efflux from the cytosol to the lumen stabilizes the position found in the hyperactive *fou2* mutant, that is, the open channel. In contrast, influx from the lumen to the cytosol drives the E605/D606 loop toward the pore, fostering channel closing. The MD simulations further suggest a $Ca^{2+}$-dependent interaction between the E605/D606 motif and a $Ca^{2+}$ coordination site at the luminal entrance of the selectivity filter (D269/E637; in VfTPC1: D271/E639). Higher luminal $Ca^{2+}$ increases the probability of $Ca^{2+}$ to occupy the coordination site which stabilized the E605/D606 motif in the pore-facing conformation. Neutralizing E605A/D606N mutations would reduce electrostatic interactions and result in channels that open with less activation energy. This effect could explain how an altered $Ca^{2+}$-binding site at the luminal pore entrance of VfTPC1 – compared with AtTPC1 – shifts the voltage activation threshold to more negative values. The difference in the voltage activation profile between AtTPC1 and VfTPC1 may have been further enhanced by a luminal $Mg^{2+}$ effect on the intact luminal pore E605/D606 motif of AtTPC1 wild type. Like $Ca^{2+}$, also luminal $Mg^{2+}$ inhibits the AtTPC1 channel but with a much lower efficiency than $Ca^{2+}$ (*Dadacz-Narloch et al., 2011*; *Pottosin et al., 2004*).

## Mutation of AtTPC1 pore motif E605/D606 does neither affect SV channel permeability nor conductance

Since the pore residues E605 and D606 of AtTPC1 are located at the luminal pore entrance relatively close to the selectivity filter (*Figure 9*, *Figure 9—figure supplement 1*; *Kintzer et al., 2018*), we investigated their contribution to the TPC1 cation permeability ($Na^+$, $K^+$, $Ca^{2+}$). The cation selectivity of AtTPC1 was first studied under bi-ionic conditions in the animal HEK cell expression system (*Guo et al., 2017*) that target TPC1 to the plasma membrane (*Figure 10A*; *Guo et al., 2016*). Therefore, for comparison and easy replacement of the bath medium at the luminal side of TPC1 channels, we also used TPC1-expressing HEK cells. To enable current recordings with AtTPC1 even under

**Table 1.** Relative cation permeability of TPC1 channel variants determined from the reversal potential under bi-ionic conditions in TPC1-expressing HEK cells.

| Channel variant | $P_K/P_{Na}$ Mean ± SD ($n$) | $P_{Ca}/P_{Na}$ Mean ± SD ($n$) |
|---|---|---|
| AtTPC1 wild type | 0.75 ± 0.08 (5)*** | - |
| AtTPC1$^{E605A/D606A}$ | 0.85 ± 0.14 (6)** | 5.43 ± 0.69 (4) |
| AtTPC1$^{D240A/D454A/E528A}$ | 0.85 ± 0.09 (5)** | 5.50 ± 0.45 (4) |
| VfTPC1 wild type | 1.18 ± 0.11 (4) | 5.92 ± 0.52 (4) |

Significant differences of $P_K/P_{Na}$ between AtTPC1 channel variants and VfTPC1 (**p < 0.01; ***p < 0.001) determined with one-way analysis of variance (ANOVA) together with a Dunnett's post hoc comparison test. The relative permeability ratios ($P_{K/ca}/P_{Na}$) obtained for AtTPC1 wild type and the AtTPC1 mutant D240A/D454A/E528A were similar to those published in *Guo et al., 2017*. Note that a similar relative permeability ratio $P_K/P_{Na}$ of 0.77 ± 0.05 (mean ± SD, $n$ = 3) was also determined from TPC1 current relaxations (*Figure 10—figure supplement 1*) recorded from wild-type mesophyll vacuoles of *Arabidopsis thaliana*.

The online version of this article includes the following source data for table 1:

**Source data 1.** Quantification of relative cation permeability of TPC1 channel variants.

high external $Ca^{2+}$ conditions (*Figure 10E*), the AtTPC1 channel mutant D240A/D454A/E528A (*Guo et al., 2017*) harboring a damaged luminal $Ca^{2+}$ sensor site 1 in the presence of a wild-type pore region was used. The relative permeability ratio $P_{K/Ca}/P_{Na}$ of AtTPC1 was not affected by disruption of $Ca^{2+}$ sensor site 3 (E605A/D606A) or $Ca^{2+}$ sensor site 1 (D240A/D454A/E528A) (*Table 1*, *Figure 10*, *Figure 10—figure supplement 1*), the latter being in well agreement with *Guo et al., 2017*. In

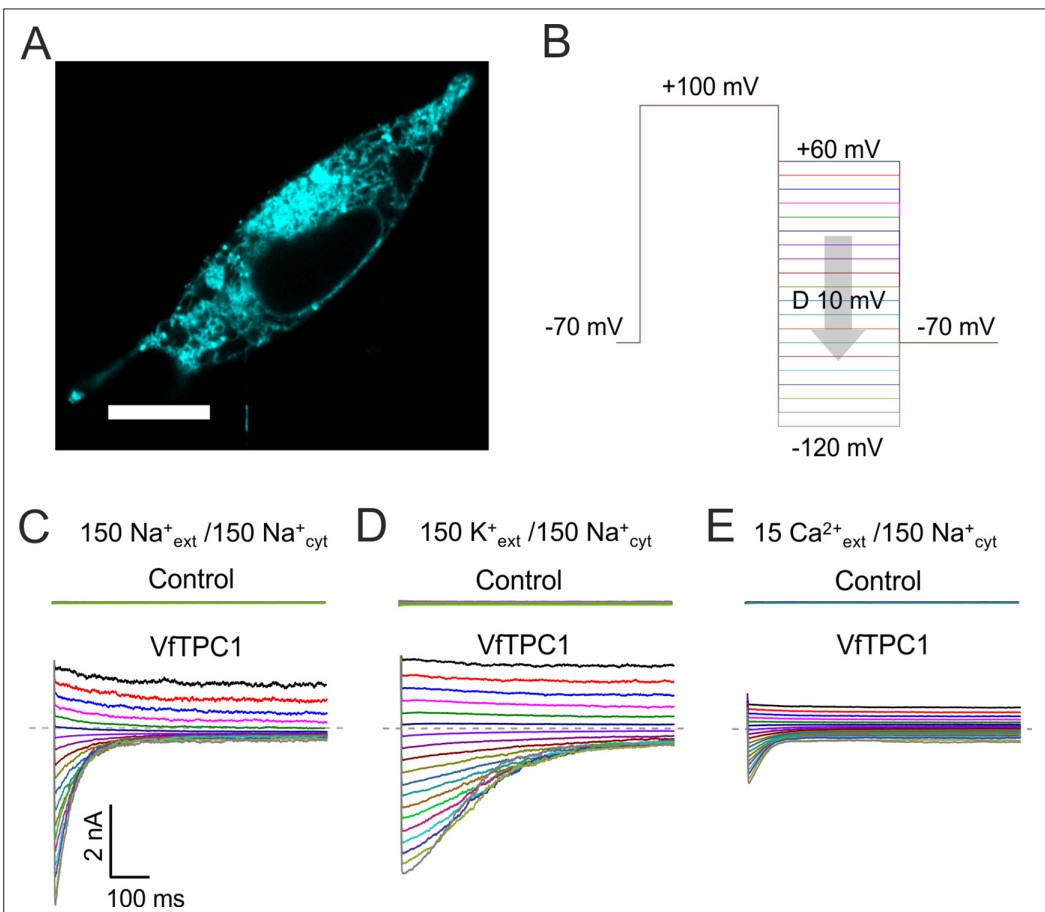

**Figure 10.** Characterization of VfTPC1 expressed in HEK293 cells. (**A**) Confocal laser scanning microscope image of a HEK293 cell expressing eYFP-VfTPC1. Note, that most fluorescence is observed in intracellular membranes but is also present in the plasma membrane. Scale bar = 10 μm. (**B–E**) Electrophysiologcial characterization of non-transfected (control) and VfTPC1-transfected HEK293 cells by whole-cell patch-clamp techniques. (**B**) Schematic overview of the protocol used during analysis including an initial activation step (−70 to +100 mV) and a subsequent voltage gradient (+60 to −120 mV) for determining the voltage dependency of TPC1. (**C–E**) Typical tail current responses of the same control or the same VfTPC1-transfected HEK cell to voltage pulses in different extracellular solutions (in mM) as indicated. The concentration of cytosolic $Na^+$ was always 150 mM. The dashed gray lines indicate the zero current level. For further details of the double-pulse protocol and patch-clamp solutions, see Materials and methods. Total number of control experiments was $n = 15$, $n = 4$, and $n = 4$ with external $Na^+$, $K^+$, and $Ca^{2+}$, respectively. Note that, on the one hand, the smaller tail currents under external $Ca^{2+}$-based conditions (**E**) compared to external $Na^+$- or $K^+$-based conditions (**C, D**) can be attributed to the 10-fold lower $Ca^{2+}$ concentration compared with $Na^+$ or $K^+$ concentration. On the other hand, these smaller tail currents under $Ca^{2+}$ conditions (**E**) also are in line with recent molecular dynamics (MD) simulations (*Navarro-Retamal et al., 2021*) suggesting that $Ca^{2+}$ cannot permeate the channel better than $K^+$ or $Na^+$. This is further supported by (1) the very small single-channel conductance under $Ca^{2+}$-based solute conditions compared to $K^+$-based solute conditions and (2) the reduced single-channel conductance at a high luminal $Ca^{2+}$ addition under $K^+$-based solute conditions (*Ward and Schroeder, 1994*; *Figure 11C*).

The online version of this article includes the following figure supplement(s) for figure 10:

**Figure supplement 1.** Electrophysiological characterization of AtTPC1 in mesophyll vacuoles of *Arabidopsis thaliana* Col0 in the whole-vacuole patch-clamp configuration.

comparison to the AtTPC1 channel variants, VfTPC1 was characterized by a similar relative cation permeability. Thus, the luminal pore motif E605/D606 does not participate in control of the TPC1 channel selectivity.

BK channels are $Ca^{2+}$-activated $K^+$ channels with large conductance, similar to TPC1 (*Barrett et al., 1982*). A negative charge ring at the pore entrance promotes the single-channel conductance of BK channels (*Brelidze et al., 2003*). To examine whether the negatively charged residues at the luminal pore entrance of AtTPC1 may play a similar role as in BK channels, single-channel currents were recorded (*Figure 11A*). Under symmetric $K^+$ conditions (100 mM) a single-channel conductance of about 80 pS was determined for AtTPC1 wild type (*Figure 11B, C*). With respect to the dimeric structure of TPC1, the neutralizations in the AtTPC1 double mutant E605A/D606A resulted in the removal of a total of four negative charges at the luminal pore entrance. Nevertheless, the unitary conductance of the mutant remained unchanged. Moreover, VfTPC1 had a threefold higher unitary conductance than AtTPC1 (250 pS, *Figure 11*; *SchulzLessdorf and Hedrich, 1995*), despite its neutral luminal pore residues at the homologous sites to AtTPC1 (E605, D606). These facts indicate that, in contrast to the AtTPC1-D269/E637 (VfTPC1-D271/E639) $Ca^{2+}$ coordination site at the luminal entrance of the selectivity filter (*Navarro-Retamal et al., 2021*), these luminal polymorphic pore residues of $Ca^{2+}$ sensor site 3 apparently do not contribute to the conductance of TPC1.

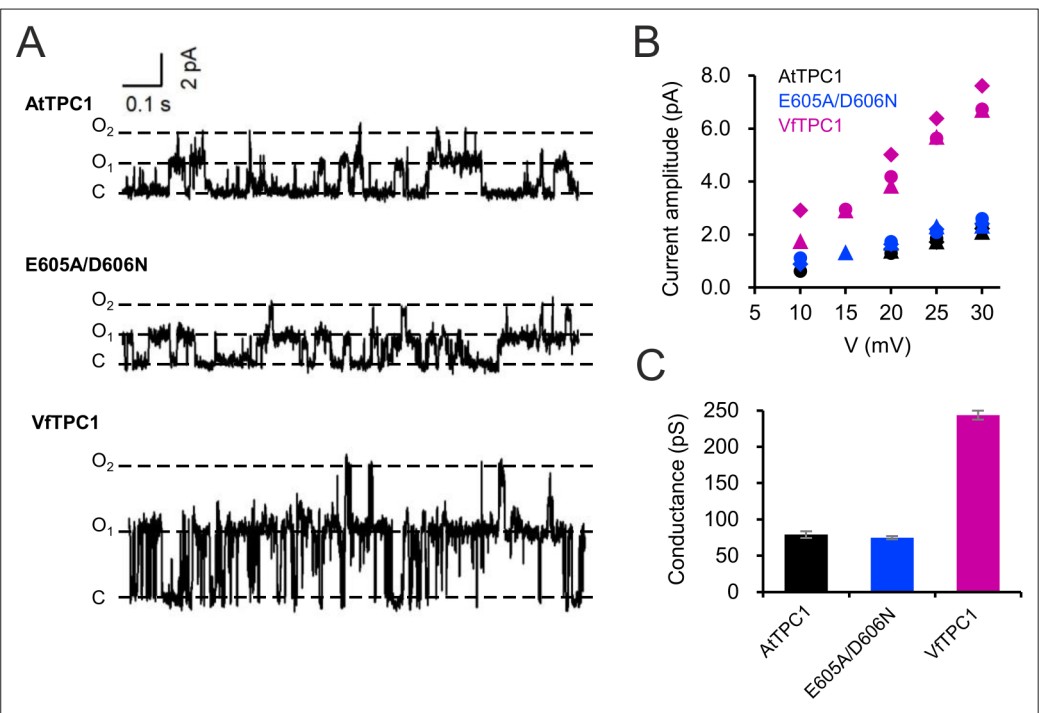

**Figure 11.** Single-channel analysis of TPC1 channel variants. (**A**) Single-channel currents of indicated TPC1 channel variants evoked at +25 mV. The current level C indicates at which all channels were closed, while at current level $O_1$ and $O_2$ one or two TPC1 channels were open. (**B**) Single-channel current amplitudes determined for indicated TPC1 channel variants plotted against the respective voltages. The channel variants are shown in different color codes. Each different symbol of the same color corresponds to an individual experiment. (**C**) Bar diagram (means ± SE) gives the unitary conductance of different TPC1 channel variants derived from the linear regression fit of the individual experiments. The number of experiments was $n$ = 3 for each channel variant. Experiments in (**A–C**) were performed with vacuoles from *attpc1-2* mesophyll protoplasts transiently transformed with the indicated TPC1 channel variants. Symmetric $K^+$ conditions (100 mM) were used with 0 and 0.5 mM $Ca^{2+}$ at the luminal and cytosolic side of the vacuole membrane, respectively. For more details on patch-clamp solutions, see Materials and methods.

The online version of this article includes the following source data for figure 11:

**Source data 1.** Quantification of single-channel conductance of TPC1 channel variants.

## Vacuoles with VfTPC1 and AtTPC1 triple mutant are hyperexcitable

The increased activity of the AtTPC1 channel mutant *fou2* leads to a hyperexcitability of the vacuole (*Jaślan et al., 2019*). Here, the membrane polarization was measured at physiological 0.2 mM luminal $Ca^{2+}$ (*Schönknecht, 2013*) with vacuoles equipped with either VfTPC1, AtTPC1 wild type or the triple AtTPC1 mutant E457N/E605A/D606N. For TPC1-dependent vacuolar excitation, depolarizing current pulses of increasing amplitudes in the range of 10–1000 pA were temporarily injected from a resting voltage of −60 mV (*Figure 12*; *Figure 12—figure supplement 1A–C*; *Figure 12—source data 1*; *Jaślan et al., 2019*). After the depolarizing current stimulus, the vacuole membrane equipped with VfTPC1 channels remained depolarized at a voltage of about 0 mV (i.e., at the equilibrium potential for $K^+$) during the entire subsequent recording period of 10 s, regardless of the stimulus intensity (*Figure 12*, *Figure 12—source data 1*). In contrast, when AtTPC1-equipped vacuoles were challenged with even the highest current stimulus (1 nA), the post-stimulus voltage remained depolarized for only a short period ($t_{plateau}$ ~ 0.4 s, *Figure 12—source data 1*) before relaxation to the resting voltage occurred. In comparison to AtTPC1 wild type, the presence of the AtTPC1 triple mutant in the vacuole membrane strongly prolonged the lifetime of the depolarized post-stimulus voltage-plateau phase at all current stimuli (*Figure 12*). Except of one vacuole, all other five AtTPC1-triple mutant vacuoles already responded to the lowest current pulses such as 10, 30, or 70 pA with sustained post-stimulus depolarization over the entire recording period (*Figure 12—source data 1*). Thus, AtTPC1-triple mutant vacuoles were also hyperexcitable, behaving very much like the VfTPC1 vacuoles. The further analysis of the corresponding TPC1 currents of these vacuoles revealed, that at this physiological luminal $Ca^{2+}$ concentration (0.2 mM), the voltage activation threshold of both, VfTPC1 and the AtTPC1 triple mutant E457N/E605A/D606N, was close to the resting membrane voltage. In comparison, AtTPC1 wild type showed a significant shift of the voltage activation threshold toward positive voltages (*Figure 12—figure supplement 1E, F*). This points to a positive correlation between the vacuolar hyperexcitability and the shifted voltage activation threshold of the hyperactive TPC1 channels (*Figure 12*, *Figure 12—figure supplement 1D–F*).

In the following we used a computational model to simulate activation of voltage-dependent TPC1-mediated vacuole excitability (*Figure 13*; *Jaślan et al., 2019*). Under standard conditions (*Figure 13*,

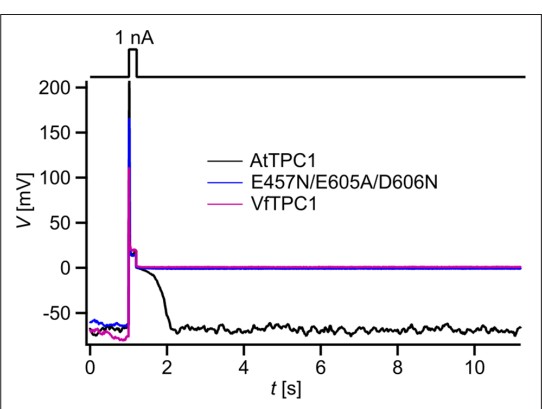

**Figure 12.** Dependency of vacuole excitability on TPC1 channel variants. Superimposed voltage responses (lower panel) of individual *attpc1-2* vacuoles equipped with either VfTPC1 wild type (magenta), AtTPC1 wild type (black), or AtTPC1-triple mutant E457N/E605A/D606N (blue) after transient transformation, to current injection of 1 nA (upper panel). Number of experiments for each channel type was $n = 6$. Lifetimes of the post-stimulus depolarization phase from each individual experiment are given in **Figure 12—source data 1**. Please note, the corresponding current responses from the same TPC1-expressing vacuoles are shown in **Figure 12—figure supplement 1D–F**. All experiments were carried out under symmetric $K^+$ conditions (150 mM) at 0.2 mM luminal free $Ca^{2+}$ and 1 mM $Ca^{2+}$ together with 2 mM $Mg^{2+}$ at the cytosolic side of the membrane. For more details on patch-clamp solutions, see Materials and methods.

The online version of this article includes the following source data and figure supplement(s) for figure 12:

**Source data 1.** TPC1-dependent lifetime of the post-stimulus depolarization phase.

**Source data 2.** Quantification of voltage- and current-clamp data shown in **Figure 12—figure supplement 1B–F**.

**Figure supplement 1.** Excitability of vacuoles with corresponding TPC1 channel activity.

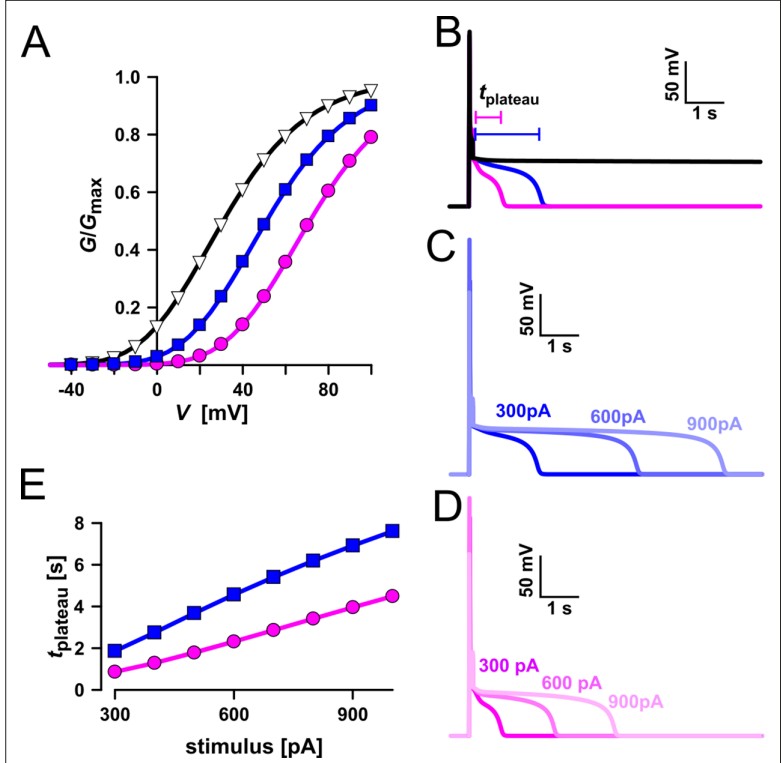

**Figure 13.** Simulation of vacuolar electrical excitability with three different TPC1 variants. (**A**) Gating characteristics ($G/G_{max}$) of three different TPC1-type channels. (**B**) Simulation of vacuolar electrical excitability with the different TPC1 variants. Excitation was induced by a 300-pA pulse of 100 ms duration. (**C**) Overlay of the electrical response of the vacuole having the blue-type TPC1 to 100 ms pulses of 300, 600, and 900 pA. (**D**) Overlay of the electrical response of the vacuole having the red-type TPC1 to 100 ms pulses of 300, 600, and 900 pA. (**E**) Dependency of the length of the post-stimulus plateau phase on the stimulus strength.

The online version of this article includes the following source data for figure 13:

**Source data 1.** Quantification of computationally simulated excitability shown in **Figure 13**.

blue) a current stimulus induced a typical excitation as observed in whole-vacuole patch-clamp experiments (**Figures 12 and 13B**). In the simulation too, the duration of the post-stimulus voltage-plateau phase ($t_{plateau}$) was found to depend on the stimulus strength (**Figure 13C, E**). The voltage activation threshold of TPC1 had an additional influence on $t_{plateau}$. A negatively shifted TPC1 activation curve (**Figure 13A**, black) provoked that the vacuole membrane remained in the excited state after stimulation (**Figure 13B**, black). A positively shifted TPC1 activation curve (**Figure 13A**, pink), however, shortened the plateau phase and thus the time course of the excited state (**Figure 13B, D**, pink). As a consequence, the slope of the linear relationship between $t_{plateau}$ and stimulus strength was reduced (**Figure 13E**). Thus, effectors regulating the voltage activation threshold of TPC1 (e.g., $Ca^{2+}$) tune the sensitivity of the vacuole membrane and shape the duration of the excited state. In comparison to AtTPC1-expressing vacuoles, those equipped with VfTPC1 or the AtTPC1 triple mutant, which both activate more negatively, are hyperexcitable (**Figure 12**).

## Discussion

We identified the fabaceae prototype VfTPC1 channel from *V. faba* as a TPC1 channel variant with gating behavior that is clearly distinct from the *A. thaliana* AtTPC1 channel of Brassicaceae. In the *Arabidopsis* background, VfTPC1 activates near vacuolar resting membrane voltage (~−30 mV; **Wang et al., 2015**) and is significantly less sensitive to luminal $Ca^{2+}$ than AtTPC1 (**Figure 5**). The dual polymorphism of two residues within the $Ca^{2+}$ sensor site 3 in the luminal pore entrance is most likely

responsible for this different gating behavior, because double loss-of-function mutations in $Ca^{2+}$ sensor site 3 of AtTPC1 resulted in VfTPC1-like AtTPC1 channels. Reverse mutations, however, did not convert the features of VfTPC1 to AtTPC1, suggesting that additional structural adaptations have evolved and counteract these efforts to gain function in terms of greater voltage and luminal $Ca^{2+}$ dependence. However, reverse gain-of-function approaches with plant ion channels also failed before (*Michard et al., 2005*; *Johansson et al., 2006*). Earlier (*Dickinson et al., 2022*), we proposed that $Ca^{2+}$ sensor site 3 in AtTPC1 is formed by three negatively charged residues (E605, D606, and D607). Here, we have now found that E605 and D606 play the most important role within this triad of site 3 in gating control in AtTPC1 (*Figure 5*). Neutral residues at this site, as naturally already present in VfTPC1 (A607/N608; *Figure 6*), uncouple luminal $Ca^{2+}$ coordination from gating (*Figure 5*).

## VfTPC1 versus AtTPC1: what is the rule and what is the exception?

*A. thaliana* is the model plant for plant molecular genetics and physiology. Is the AtTPC1 structure–function relation reference for all plant SV channels? Obviously not! In comparison to AtTPC1, VfTPC1 is hyperactive and less sensitive toward luminal $Ca^{2+}$ (*Figures 2 and 5* and *Figure 2—figure supplement 1*). In fact, these VfTPC1 features may be even more widespread in the plant kingdom, because TPC1 channels from some other plant species like for example *Lotus japonicus*, *Oryza sativa*, and *Hordeum vulgare* also exhibit the neutralizing triple-polymorphism characteristic of VfTPC1 in the luminal $Ca^{2+}$ sensor sites 2 and 3 (*Figures 6 and 8*). In well agreement with this triple polymorphism, the fabacean TPC1 channel from *L. japonicus* shows a VfTPC1-like gating behavior and low luminal $Ca^{2+}$ sensitivity as TPC1 currents can still be recorded at 50 mM luminal $Ca^{2+}$ (see *Appendix 1—figure 1*). Similarly, voltage- and $Ca^{2+}$-dependent TPC1 current responses from mesophyll vacuoles of the monocots *O. sativa* and *H. vulgare* appear also to exhibit a lower luminal $Ca^{2+}$ sensitivity and higher probability to open at more negative voltages, respectively (*Dadacz-Narloch et al., 2013*; *Lenglet et al., 2017*; *Pottosin et al., 1997*). Such luminal pore polymorphism, feeding back on gating behavior of TPC1 channels, could allow species to adapt the electrical properties of the vacuole to given individual environmental settings. In support of this hypothesis population-based evolutionary genomic approaches have pinpointed a polymorphic pore residue – other than the E605/D606 motif – in the vicinity of the TPC1 selectivity filter of *Arabidopsis arenosa* populations adapted to non-serpentine and serpentine soils (*Arnold et al., 2016*; *Konečná et al., 2021*; *Turner et al., 2010*). Serpentine barrens are characterized by skewed $Ca^{2+}/Mg^{2+}$ ratios and elevated heavy metals, often combined with low nutrient availability and drought conditions. While the functional consequences of this polymorphism for ion permeation remain elusive and subject to future studies, these results suggest that TPC1 channels are also involved in control of ion homeostasis in addition to a possible contribution to the vacuolar $Ca^{2+}$ homeostat (*Dindas et al., 2021*; see below). Of note, this residue is also polymorphic in the legume clade of TPC1 channels. In legumes, rhizobia-triggered nodule formation depends on $Ca^{2+}$ signals and is accompanied by a pronounced $K^+$ loss from the vacuole during the life span of infected cells (*Fedorova et al., 2021*). It is tempting to speculate that the peculiar luminal $Ca^{2+}$ tolerance of legume TPC1 type promote nodule formation and control ion homeostasis during nodule development. This hypothesis can be tested by generating TPC1-loss-of-function mutants in the model legumes Lotus and Medicago.

## The complex relationship between TPC1 and $Ca^{2+}$

In well agreement with experiments with *V. faba* vacuoles (*Ivashikina and Hedrich, 2005*; *Ward and Schroeder, 1994*), our cation replacement experiments with TPC1-expressing HEK cells demonstrate the principal ability of VfTPC1 and AtTPC1 to conduct not only $K^+$ and $Na^+$ but also $Ca^{2+}$ ions. Although relative permeability ratios of $P_{Ca}/P_{Na} \approx 5:1$ (*Table 1*) may suggest a high calcium permeability, the reduced single-channel conductance measured with $Ca^{2+}$ (*Ward and Schroeder, 1994*) support the recent mechanistic explanation that relative permeabilities of TPC1s do not correlate with the actual $Ca^{2+}$ conductivity (*Navarro-Retamal et al., 2021*). Under physiological, plant cell-like $Ca^{2+}$ and $K^+$ concentrations and gradients, the fava bean SV/TPC1 channel was predominately conducting $K^+$ (*SchulzLessdorf and Hedrich, 1995*, for review see *Hedrich and Marten, 2011*; *Hedrich et al., 2018*; *Pottosin and Dobrovinskaya, 2022*). This permeation profile is consistent with the $K^+$-starvation transcriptome and the elevated vacuolar $Ca^{2+}$ content of *Arabidopsis* leaves equipped with hyperactive *fou2* TPC1 channels (*Beyhl et al., 2009*; *Bonaventure et al., 2007b*). The latter suggests that the

luminal $Ca^{2+}$-binding sites of TPC1 or, in other words, the susceptibility of TPC1 to block by elevated luminal $Ca^{2+}$ may be the key for the $Ca^{2+}$ storage capacity of the plant vacuole. At the same time, a much lower luminal $Ca^{2+}$ sensitivity would then keep TPC1 functional even at high luminal vacuolar $Ca^{2+}$ levels. Based on the observation that TPC1/SV channels are in principle permeable to $Ca^{2+}$ under certain experimental conditions, it was speculated that TPC channels (similar to the animal field) are part of a $Ca^{2+}$-induced $Ca^{2+}$ release (CICR) network (*Ward and Schroeder, 1994*). Intriguingly, studies on TPC1 overexpressing and loss-of-function *Arabidopsis* mutants showed that the AtTPC1 activity modulates the speed and not the amplitude of traveling $Ca^{2+}$ waves in roots triggered by salt stress (*Choi et al., 2014*). Quantitative analysis of local and distant $Ca^{2+}$ waves in leaves elicited by wounding, however, excluded the role of TPC1 channels as a CICR element in the vacuole membrane (*Bellandi et al., 2022*). Alternatively, TPC1-triggered vacuole excitation could cause the release of $Ca^{2+}$ from the vacuole to the cytosol via a $Ca^{2+}$ homeostat (i.e., a combination of yet-to-be identified $Ca^{2+}$ transporters, *Dreyer, 2021*) with a striking correlation between the duration of excitation and the amount of released $Ca^{2+}$ (*Dindas et al., 2021*). In such a model scenario, the control of the voltage activation threshold of TPC1 by luminal $Ca^{2+}$ (*Figure 5*) would directly influence the excitation duration of the vacuole membrane (*Figures 12 and 13E*) and $Ca^{2+}$ release in turn. Without such a regulatory process, a higher luminal $Ca^{2+}$ concentration would imply a larger transmembrane $Ca^{2+}$ gradient and therefore the release of a larger $Ca^{2+}$ quantum. Tuning of the TPC1 activity by luminal $Ca^{2+}$ would thus have the potential to mitigate or even counterbalance the electro-chemical potential of a larger $Ca^{2+}$ gradient. A higher luminal $Ca^{2+}$ concentration downregulates TPC1 activity, which in turn would reduce the excitation duration and would result in a shortened $Ca^{2+}$ release of larger amplitude. Since TPC1 channels are involved in certain $Ca^{2+}$-dependent stress responses (*Pottosin and Dobrovinskaya, 2022*), such a scenario could be triggered by a cytosolic $Ca^{2+}$ signal mediated by $Ca^{2+}$-permeable channels of the plasma membrane or ER membrane. Given the involvement of TPK channels to vacuolar excitation (*Jaślan et al., 2019*), the increased cytosolic $Ca^{2+}$ level could initially lead to the activation of TPK channels for initial depolarization of the vacuolar membrane. Consistent with the findings on the TPC1 activation sequence from recent molecular TPC1 structures (*Dickinson et al., 2022*; *Ye et al., 2021*), these two signals, voltage together with cytosolic $Ca^{2+}$, could activate TPC1 channels and lead to further depolarization (*Allen and Sanders, 1996*) affecting the vacuolar $Ca^{2+}$ homeostat. The TPC1-dependent activation of a vacuolar $Ca^{2+}$ homeostat may also explain the wounding and retarded growth phenotype of the *fou2* mutant (*Bonaventure et al., 2007a*) in comparison to *Arabidopsis* wild type. The presented model scenario could also be imagined for *V. faba*. Nevertheless, the absence of a retarded growth phenotype of these plants may point to species-specific differences. In view of the high impact of the TPC1 activation threshold on hormone homeostasis and plant growth, future research is now warranted on how *V. faba* plants can (1) prevent retarded growth and (2) benefit from a TPC1 channel variant with a greatly reduced luminal $Ca^{2+}$ sensitivity for potentially better adaptation to their respective ecological niche.

## Materials and methods

### Plant materials and growth conditions

*A. thaliana* wild type (Col0) and *attpc1-2* mutant (*Peiter et al., 2005*) were grown in a growth chamber under short day conditions (8 hr light, 16 hr dark) with a day/night temperature regime of 22/16°C, a photon flux density of 150 µmol m$^{-2}$ s$^{-1}$ and a relative humidity of about 60%. *V. faba* and *L. japonicus* plants were grown in the greenhouse at a 16-hr day/8-hr dark photoperiod, a day/night temperature regime of 22/18°C and a light intensity of approximately 1250 µmol m$^{-2}$ s$^{-1}$.

### RNA sequencing

Leaves from 6- to 8-week-old *V. faba* plants were harvested and mesophyll RNA was extracted using the NucleoSpin Plant RNA extraction kit (Macherey-Nagel, Düren, Germany) according to the manufacturer's instructions. For guard cell RNA extraction epidermal fragments were isolated using the blender method (*Bauer et al., 2013*; *Raschke and Hedrich, 1989*). Total RNA from three individual biological replicates was prepared and subjected to RNA-sequencing on an Illumina NextSeq500 platform. High-quality RNA-seq paired-end reads were quality checked using FastQC (version 0.11.6) and transcriptomes were de novo assembled individually using Trinity (version 2.5.1 Release). Finally, the

TRAPID pipeline was employed for processing of assembled transcriptome data including transcript annotation (*Bucchini et al., 2021*). Based on AtTPC1 homology, identical VfTPC1 transcripts were identified in mesophyll and guard cell fractions, and the obtained sequence information was used to clone the VfTPC1 CDS by a PCR-based approach. The *VfTPC1* mRNA sequence is deposited at GenBank under the following accession number: VfTPC1_mRNA MW380418.

## Cloning and site-directed mutagenesis

After the total RNA was extracted from mature leaves of *V. faba* and *L. japonicus* plants and reverse transcribed into complementary DNA (cDNA), VfTPC1 and LjTPC1 were amplified from the cDNA libraries, essentially as described by *Dadacz-Narloch et al., 2013*. For patch-clamp experiments with vacuoles, the cDNA coding sequences of the AtTPC1 channel variants were cloned into the modified pSAT6-eGFP-C1 vector (GenBank AY818377.1), whereas the coding sequences of the different VfTPC1 and LjTPC1 channel variants were cloned into the pSAT6-eYFP-C1 vector (GenBank DQ005469) using the uracil-excision-based cloning technique (*Nour-Eldin et al., 2010*), essentially as described by *Dadacz-Narloch et al., 2011*. The resulting TPC1-eGFP constructs were under the control of the 35 S promoter, and the TPC1-eYFP constructs were under the control of the ubiquitin promoter (UBQ10). For patch-clamp experiments with HEK cells, the eYFP coding sequence (*Nagai et al., 2002*) was fused without the stop codon to the 5′ end of the TPC1 cDNA coding sequences and then cloned together into the pcDNA3.1 vector (GenBank MN996867.1). In analogy to *Dadacz-Narloch et al., 2011*, a modified USER fusion method was used to introduce site-directed mutations in the wild-type AtTPC1 construct. The sequences of the primers used for subcloning and mutagenesis are listed in *Supplementary file 2*. All channel variants were tested for their sequences.

## Transient protoplast transformation

Essentially following the protocols from *Sheen, 2002* and *Yoo et al., 2007*, mesophyll protoplasts were released from 6- to 7-week-old *tpc1-2 Arabidopsis* mutant plants and transiently transformed with the different TPC1 channel constructs. For channel expression, protoplasts were then stored in W5 solution (125 mM CaCl$_2$, 154 mM NaCl, 5 mM KCl, 5 mM glucose, 2 mM MES (2-(N-Morpholino) ethanesulfonic acid)/Tris, pH 5.6, 50 μg ml$^{-1}$ ampicillin) at 23°C in the dark for usually 2 days.

## HEK cell transfection

The 293 embryonic renal epithelial cell line authenticated by STR profiling was acquired from ATCC (American Type Culture Collection, Manassas, VA, USA). During laboratory research use the cell line identity has been authenticated by microscopy and the cell culture was negatively tested for mycoplasms by Eurofins (Ebersberg, Germany). HEK293 cells were transfected with the respective eYFP-TPC1-fused constructs with Lipofectamine 2000 (Thermo Fisher, Waltham, USA) according to the manufacturer's instructions. Cells were seeded 18–24 hr after transfection on glass coverslips (diameter 12 mm). Protein expression was verified on the single-cell level by the appearance of eYFP fluorescence in the plasma membrane upon excitation with a 473-nm laser.

## Subcellular targeting

Vacuolar membrane localization of the expressed eGFP/eYFP-fused TPC1 channels was verified by imaging the fluorescence signal of transformed protoplasts and vacuoles with a confocal laser scanning microscope (TCS SP5, Leica, Mannheim, Germany) (*Dadacz-Narloch et al., 2013*). eGFP and eYFP were excited with an Argon laser at 490 and 514 nm, respectively, and the emission of fluorescence was monitored between 500 and 520 nm for eGFP and between 520 and 540 nm for eYFP. Red autofluorescence of chlorophyll was excited at 540 nm and acquired between 590 and 610 nm. For expression analysis in HEK293 cells, a confocal laser scanning microscope (SP700, Zeiss, Germany) equipped with three laser lines (488 nm: 10 mW, 555 nm: 10 mW, 639 nm: 5 mW) was used. Images were processed with ZEN software (ZEN 2012, Zeiss) or Fiji, Version ImageJ 1.50 (*Schindelin et al., 2012*).

## Whole-vacuole patch-clamp experiments

Mesophyll vacuoles were liberated from *A. thaliana* wild-type protoplasts after their enzymatical isolation, essentially as described in *Beyhl et al., 2009*. Vacuoles were released from mesophyll

protoplasts of the *attpc1-2* mutant in the recording chamber 2 days after transformation using a vacuole release (VR) solution (*Lagostena et al., 2017*). The VR solution was modified and composed of 100 mM malic acid, 155 mM NMDG (*N*-methyl-D-glucamine), 5 mM EGTA (ethylene glycol-bis(β-aminoethyl ether)-N,N,N',N'-tetraacetic acid), 3 mM $MgCl_2$, 10 mM HEPES(4-[2-hydroxyethyl] piperazine-1-ethanesulfonic acid)/Tris pH 7.5 and adjusted to 450 mOsmol $kg^{-1}$ with D-sorbitol. The whole-vacuole configuration was then established either with wild-type vacuoles or TPC1-transformed vacuoles. The latter vacuoles were easily identified upon their eGFP- or eYFP-based fluorescence measured between 510 and 535 nm after excitation at 480 nm with a *precisExcite HighPower* LED lamp (Visitron Systems GmbH, Puchheim, Germany). Patch pipettes were prepared from Harvard glass capillaries (Harvard glass capillaries GC150T-10, Harvard Apparatus, UK) and typically had a resistance in the range of 1.4–3.1 MΩ. The membrane capacitance ($C_m$) of the individual vacuoles accessed and compensated in patch-clamp experiments ranged from 31.1 to 68.5 pF. When the seal was stable throughout the experiment, membrane currents and voltages were recorded with a sampling rate of 150 μs at a low-pass filter frequency of 2.9 or 3 kHz using an EPC10 or EPC800 patch-clamp amplifier, respectively (HEKA Electronic). Data were acquired with the software programs Pulse or Patchmaster (HEKA Electronic) and offline analyzed with IGORPro (Wave Metrics).

Voltage recordings were carried out with TPC1-transformed vacuoles in the current-clamp mode as described by *Jaślan et al., 2019*. Briefly, after adjusting the membrane voltage to −60 mV by injection of an appropriate current, current pulses were applied in the range of 10–1000 pA for 200 ms. In the voltage-clamp experiments, macroscopic currents were recorded in response to 1-s-lasting voltage pulses in the range of −80 to +110 mV in 10 mV increments applied from a holding voltage of −60 mV. Following each voltage pulse, the holding voltage of −60 mV was applied again. The total time interval between voltage pulses at holding voltage was 4 s. The corresponding current responses of each vacuole were analyzed with respect to the half-activation time ($t_{act-0.5}$) and steady-state current amplitudes ($I_{ss}$). The half-activation time ($t_{act-0.5}$) was the time at which 50% of the steady-state current amplitude was reached. As a normalization measure for the membrane surface of the individual vacuole, the determined steady-state currents were divided by the respective compensated membrane capacitance ($C_m$). Conductance/voltage curves ($G/G_{max}(V)$) were quantified from tail current experiments as a measure for the relative voltage-dependent open-channel probability. Following pre-pulse voltages in the range of −80 to +110 mV, instantaneous tail currents were determined at −60 mV. The midpoint voltages ($V_1$, $V_2$) and the equivalent gating charges ($z_1$, $z_2$) were derived by fitting the $G/G_{max}(V)$ curves with a double Boltzmann equation (*Dickinson et al., 2022*). After pre-activation of TPC1 currents upon a 300-ms-lasting instantaneous voltage pulse to either +80 mV (VfTPC1) or +100 mV (AtTPC1 channel variants), current relaxation was recorded for 200 ms at voltages ranging from −60 to 0 mV in 10 mV decrements. Before and after the double-voltage pulse, the membrane was clamped to the holding voltage of −60 mV. The total time interval between the double-voltage pulses at holding voltage was 4.3 s. The deactivation-half times ($t_{deact-0.5}$) were determined at which the initial tail current amplitude has declined by 50%.

In the voltage-clamp experiments of *Figures 2–5* and *Figure 7*, *Figure 2—figure supplement 1*, *Appendix 1—figure 1*, the standard bath solution facing the cytoplasmic side of the vacuole membrane contained 150 mM KCl, 1 mM $CaCl_2$, 10 mM HEPES (pH 7.5/Tris) and was adjusted with D-sorbitol to an osmolality of 520 mOsmol $kg^{-1}$. The standard pipette solution at the luminal side of the tonoplast basically consisted of 150 mM KCl, 2 mM $MgCl_2$, 10 mM HEPES (pH 7.5/Tris) and was adjusted with D-sorbitol to a final osmolality 500 mOsmol $kg^{-1}$. The pipette solution was supplemented with 10 or 50 mM $CaCl_2$ or adjusted to 0 mM $Ca^2$ by addition of 0.1 mM EGTA. Current-clamp experiments (*Figure 12*, *Figure 12—figure supplement 1A–C*) and corresponding voltage-clamp experiments (*Figure 12—figure supplement 1D–F*) were performed on the same individual vacuoles. For these measurements, the standard bath medium additionally contained 2 mM $MgCl_2$ and the pipette solution was adjusted to 0.2 mM free $Ca^{2+}$ by addition of 4.1 mM EGTA and 4.3 mM $Ca^{2+}$ (maxchelator/webmaxc/webmaxcS.htm). It should be noted that voltage- and current-clamp experiments with non-transformed *A. thaliana attpc1-2* vacuoles as negative controls have been published previously by *Jaślan et al., 2019*; *Jaślan et al., 2016*, respectively.

## Patch-clamp experiments with membrane patches

Vacuoles were isolated from transiently transformed protoplasts by perfusion with a solution containing 10 mM EGTA, 10 mM HEPES/Tris pH 7,5, adjusted to 200 mOsmol kg$^{-1}$ with D-sorbitol. Excised membrane patches with the cytosolic side of the tonoplast facing the bath medium were formed from the whole-vacuole configuration. Single-channel fluctuations were recorded with a sampling rate of 100 µs at a low-pass filter frequency of 1 kHz using an EPC10 patch-clamp amplifier. Single-channel current amplitudes were determined from all-point histograms of the current recordings and plotted against the respective voltages. The single-channel conductance for each membrane patch was derived from linear regression of the current–voltage plot. The bath medium contained 100 mM KCl, 0.5 mM CaCl$_2$, 10 mM HEPES (pH 7.5/Tris). The pipette medium consisted of 100 mM KCl, 2 mM MgCl$_2$, 2 mM EGTA, 10 mM MES (pH 5.5/Tris). Both solutions were adjusted to 400 mOsmol kg$^{-1}$ with sorbitol.

## Patch-clamp experiments with HEK cells

Whole-cell current recordings were performed at a setup described previously (*Panzer et al., 2019*). Data were acquired using Clampex 10.7 (Molecular devices, San Jose, USA) with 100 kHz sampling rate, low-pass filtered at 5 kHz, and analyzed with Clampfit 10.7 (Molecular devices) and OriginPro 2016 (Originlab, Northampton, USA). Pipette resistance (GB150F-8P, Scientific-Instruments, Hofheim, Germany) was 4–6 MΩ in standard bath solution. The standard pipette solution contained 150 mM NaCl, 2.5 mM MgCl$_2$, 0.3 mM free Ca$^{2+}$ (adjusted by 4.3 mM CaCl$_2$ and 4 mM EGTA), 10 mM HEPES (pH 7.4/Tris). The standard bath solution was composed of 150 mM NaCl, 10 mM HEPES (pH 7.4/Tris). Under these experimental conditions bath and pipette solutions mimicked the vacuolar and cytosolic solute conditions, respectively. To determine the relative permeability ratio on the same HEK cell under bi-ionic cation conditions, the Na$^+$-based bath medium was replaced by either a K$^+$- or Ca$^{2+}$-based solution. The K$^+$-based bath medium contained 150 mM KCl, 10 mM HEPES (pH 7.4/Tris), and the Ca$^{2+}$-based one consisted of 15 mM CaCl$_2$, 120 mM NMDG-Cl, 10 mM HEPES (pH 7.4/Tris). Only cells were included in the analysis that exhibited stable seal conditions throughout the whole set of solution exchange.

For determination of the TPC1 ion selectivity, tail current experiments were conducted. After pre-activating the TPC1 channels by a voltage step from −70 mV (resting potential) to +100 mV for 500–1000 ms, the relaxation of the outward currents in response to hyperpolarizing voltage pulses was recorded either with 10 mV intervals reaching from +60 to −120 mV or in 5 mV decrements reaching from +30 to −40 mV to visualize and to analyze the reversal potentials, respectively. After the double-voltage pulse, the membrane was clamped again to the holding voltage of −70 mV. The total time interval between the double-voltage pulses at holding voltage was 3.4 s. In order to clearly assign the reversal potential to TPC1 currents, which are not contaminated by leakage currents, the slope of the tail currents was determined and plotted against the corresponding voltages. The reversal potential for each solute condition was then determined by interpolation the smallest negative and positive slope values. The shift in the reversal potential $\Delta V_{rev}$ caused by the change of the external solution from either Na$^+$ to K$^+$ or Na$^+$ to Ca$^{2+}$ solution was used to estimate the relative permeability ratios $P_{Ca}/P_{Na}$ or $P_K/P_{Na}$ essentially as described by *Sun et al., 1997*. For calculation of the K$^+$ to Na$^+$ permeability ratio ($P_K/P_{Na}$) the following *Equation 1* was used (*Hille, 1971*):

$$\frac{P_K}{P_{Na}} = \frac{[K]}{[Na]} \cdot \exp\left(\frac{\triangle V_{rev}}{\alpha}\right) \tag{1}$$

where $\Delta V_{rev} = V_K - V_{Na}$, with the reversal potentials $V_K$ and $V_{Na}$ in K$^+$- or Na$^+$-based bath solution, respectively, and $\alpha = RT/F = 25.42$ mV at 22°C. The relative permeability ratio $P_{Ca}/P_{Na}$ was calculated using *Equation 2* derived from the extended Goldman–Hodgkin–Katz equation (*Lewis, 1979*):

$$\frac{P_{Ca}}{P_{Na}} = \frac{\left(1 + \exp\left(\frac{V_{Ca}}{\alpha}\right)\right) \cdot \exp\left(\frac{\triangle V_{rev}}{\alpha}\right) \cdot [Na]}{4 \cdot [Ca]} \tag{2}$$

where $\Delta V_{rev} = V_{Ca} - V_{Na}$ with the reversal potentials $V_{Ca}$ and $V_{Na}$ in Ca$^{2+}$- or Na$^+$-based bath solution, respectively, and $\alpha$ has same meaning as above.

### Voltage convention in patch-clamp experiments

The given membrane voltages refer to the cytosolic side of the vacuole membrane or HEK cell plasma membrane with zero potential on its luminal or extracellular side, respectively. In experiments with vacuoles performed in the presence of 50 mM luminal $Ca^{2+}$ (*Figures 2–5*, and *Appendix 1—figure 1*), the membrane voltages were corrected offline by the corresponding liquid junction potential determined offline according to *Neher, 1992* and *Ward and Schroeder, 1994*. Otherwise, no correction for the liquid junction potential was necessary.

### Statistical analysis

Patch-clamp experiments were conducted with individual vacuoles of *attpc1-2* mutant plants, transformed with a specific channel construct, or wild-type plants under a given solute condition. Each individual vacuole patch-clamp experiment represents a biological replicate, that is the mesophyll vacuoles studied were derived from different plants. Experiments with individual HEK293 cells yielded from the same or different transfections were considered technical and biological replicates, respectively (*Table 1—source data 1*). Due to a common dataset, $I_{ss}/C_m(V)$ and $G/G_{max}(V)$ curves for wild-type AtTPC1 acquired under 0 and 10 mM luminal $Ca^{2+}$ were identical to those shown in *Dickinson et al., 2022*. Electrophysiological data are given as means ± standard error or standard deviation as indicated in the figure legends or tables. The statistical analysis was performed using one-way analysis of variance (ANOVA) followed by the Dunnett's post hoc comparison test. In order to test also the $Ca^{2+}$-induced shifts in the $V_{1/2}$ values for significant differences, the $V_1$ and $V_2$ mean values determined in the absence of luminal $Ca^{2+}$ were subtracted from the individual $V_1$ and $V_2$ values, respectively, derived under 10 mM luminal $Ca^{2+}$ conditions (*Dickinson et al., 2022*). In analogy, the $Ca^{2+}$-induced change in the $t_{act-0.5}$ values were analyzed for significant differences. Statistical analysis was done with Origin 2020 (OriginLab, Northampton, MA, USA) and SPSS 2020 (IBM, New York, USA).

### 3D modeling

An atomic model of VfTPC1 was generated using the homology modeling program MODELLER B (*Webb and Sali, 2016*), using the experimental AtTPC1 D454N (*fou2*) $Ca^{2+}$-bound structure (*Dickinson et al., 2022*) as a reference (PDB: 7TBG). The model was then relaxed into the 2.5 Å resolution *fou2*-$Ca^{2+}$ map (*Dickinson et al., 2022*) using ISOLDE (*Croll, 2018*) and used for atomic interpretation.

### In silico experiments

Electrical excitability of the vacuolar membrane was computationally simulated as described in detail before (*Jaślan et al., 2019*) involving a background conductance, voltage-independent $K^+$ channels of the TPK-type and the time- and voltage-dependent cation channel TPC1 that confers excitability to the vacuolar membrane. Details are also provided in *Supplementary file 3*.

## Acknowledgements

We are grateful to Parathy Yogendran for transient transfection of HEK293 cells, Matthias Freund and Arthur Korte for support in statistical analysis and transcriptome assembly, respectively, and Armando Carpaneto for discussion. We also thank Heike M. Müller for support in RNA preparation.

## Additional information

#### Funding

| Funder | Grant reference number | Author |
| --- | --- | --- |
| Deutsche Forschungsgemeinschaft | Koselleck award HE 1640/42-1 | Rainer Hedrich |
| Deutsche Forschungsgemeinschaft | priority programs 'MAdLand - Molecular Adaptation to Land: Plant Evolution to Change' HE 1640/45-1 | Rainer Hedrich |

| Funder | Grant reference number | Author |
|---|---|---|
| Deutsche Forschungsgemeinschaft | priority programs 'MAdLand - Molecular Adaptation to Land: Plant Evolution to Change' BE1867/9-1 | Dirk Becker |
| China Scholarship Council | doctoral fellowship | Jinping Lu |
| Deutscher Akademischer Austauschdienst | STIPET fellowship | Jinping Lu |
| Comisión Nacional de Investigación Científica y Tecnológica | FONDEQUIP EQM160063 | Ingo Dreyer |
| Anilio-Anid | ATE220043 | Ingo Dreyer |
| Fondo Nacional de Desarrollo Científico y Tecnológico | 3170434 | Carlos Navarro-Retamal |
| Fondo Nacional de Desarrollo Científico y Tecnológico | 1220504 | Ingo Dreyer |

The funders had no role in study design, data collection, and interpretation, or the decision to submit the work for publication.

## Author contributions

Jinping Lu, Performed cloning and site-directed mutagenesis, transient protoplast transformation, subcellular targeting experiments, patch clamp experiments with vacuoles and analyzed corresponding data, assisted with preparing the manuscript; Ingo Dreyer, Designed and analyzed the in silico experiments, wrote the manuscript; Miles Sasha Dickinson, Performed VfTPC1 homology modeling, wrote the manuscript; Sabine Panzer, Performed HEK cell experiments; Dawid Jaślan, Assisted in patch clamp experiments on vacuoles; Carlos Navarro-Retamal, Conducted and analyzed the in silico experiments; Dietmar Geiger, Assisted with cloning and site-directed mutagenesis; Ulrich Terpitz, Performed HEK cell experiments, analyzed corresponding data, assisted with preparing the manuscript; Dirk Becker, Performed the consensus tree, transcriptome assembly and analysis, used the VfTPC1 homology model for further visualization, wrote the manuscript; Robert M Stroud, Designed the work for homology modeling, wrote the manuscript; Irene Marten, Designed the electrophysiological research, supervised the project, wrote the manuscript; Rainer Hedrich, Designed the electrophysiological research, wrote the manuscript

## Author ORCIDs

Jinping Lu https://orcid.org/0000-0003-1632-0597
Ingo Dreyer https://orcid.org/0000-0002-2781-0359
Sabine Panzer https://orcid.org/0000-0002-8476-3595
Dawid Jaślan https://orcid.org/0000-0002-0685-7633
Carlos Navarro-Retamal https://orcid.org/0000-0001-9581-0206
Dietmar Geiger https://orcid.org/0000-0003-0715-5710
Ulrich Terpitz https://orcid.org/0000-0003-3031-3422
Dirk Becker https://orcid.org/0000-0003-0723-4160
Irene Marten https://orcid.org/0000-0001-8402-869X
Rainer Hedrich https://orcid.org/0000-0003-3224-1362

## Decision letter and Author response

Decision letter https://doi.org/10.7554/eLife.86384.sa1
Author response https://doi.org/10.7554/eLife.86384.sa2

# Additional files

## Supplementary files

• Supplementary file 1. Homology model of *Vicia faba* TPC1 based on the experimental AtTPC1

D454N (*fou2*) Ca²⁺-bound structure (*Dickinson et al., 2022*; PDB: 7TBG).

- Supplementary file 2. Primer sequences for subcloning and side-directed mutagenesis.
- Supplementary file 3. Details of the method used for computational simulation of electrical excitability shown in *Figure 13*.
- MDAR checklist

## Data availability

All data are included in the manuscript and supporting materials. Source data files were provided for the *Figures 2–7 and 11*, *Appendix 1—figure 1*, and *Table 1*; they contain numerical data to generate the figures. New sequencing data have been deposited at Genbank under the accession number VfTPC1_mRNA MW380418. Published protein structures were used for protein modeling in *Figure 9* (PDB: 7TBG, 5DQQ) and *Figure 9—figure supplement 1* (PDB: 5TUA). The homology model of VfTPC1 was provided by *Supplementary file 1*. For direct comparison, a small electrophysiological dataset published by *Dickinson et al., 2022* were reused as indicated in the corresponding figure legends (*Figures 2 and 5*).

The following dataset was generated:

| Author(s) | Year | Dataset title | Dataset URL | Database and Identifier |
|---|---|---|---|---|
| Lu J, Dreyer I, Dickinson MS, Panzer S, Jaslan D, Navarro-Retamal C, Geiger D, Terpitz U, Becker D, Stroud RM, Marten I, Hedrich R | 2022 | Vicia faba var. faba vacuolar two pore Ca2+-channel (TPC1) mRNA, complete cds | https://www.ncbi.nlm.nih.gov/nuccore/MW380418 | NCBI Nucleotide, MW380418 |

The following previously published datasets were used:

| Author(s) | Year | Dataset title | Dataset URL | Database and Identifier |
|---|---|---|---|---|
| Becker D, Marten I, Hedrich R | 2020 | Vicia faba var. faba vacuolar two pore Ca2+-channel (TPC1) mRNA, complete cds | https://www.ncbi.nlm.nih.gov/nuccore/2258117146 | NCBI GenBank, 2258117146 |
| Kintzer AF, Stroud RM | 2015 | Structure, inhibition and regulation of two-pore channel TPC1 from *Arabidopsis thaliana* | https://www.rcsb.org/structure/5DQQ | RCSB Protein Data Bank, 5DQQ |
| Guo J, Zeng W, Jiang Y | 2016 | Structure of a Na+-selective mutant of two-pore channel from *Arabidopsis thaliana* AtTPC1 | https://www.rcsb.org/structure/5TUA | RCSB Protein Data Bank, 5TUA |
| Dickinson MS, Stroud RM | 2021 | AtTPC1 D454N with 1 mM Ca2+ | https://www.rcsb.org/structure/7TBG | RCSB Protein Data Bank, 7TBG |

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

## Appendix 1

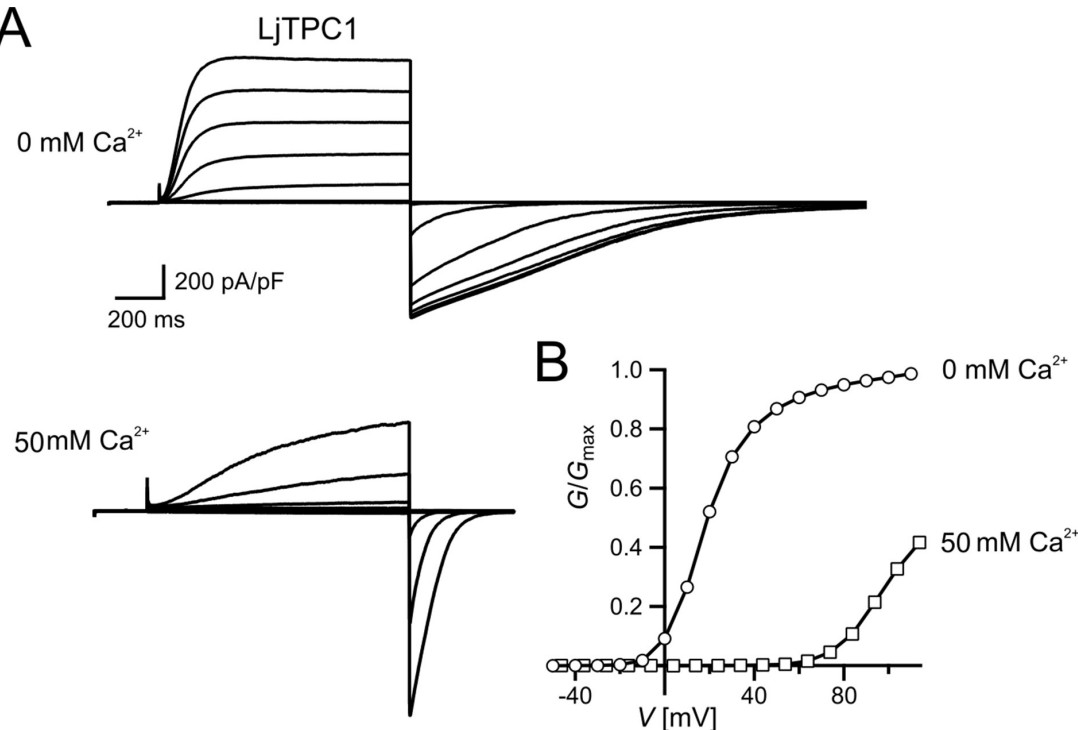

**Appendix 1—figure 1.** Similar voltage-dependent gating behavior of TPC1 from *Lotus japonicus.* (**A**) Macroscopic TPC1 current recordings from individual mesophyll vacuoles at 0 and 50 mM luminal Ca²⁺ after transient transformation of *Arabidopsis thaliana* mesophyll protoplasts (attpc1-2) with LjTPC1 from *L. japonicus* TPC1 currents. Currents were elicited upon depolarizing voltages pulses in the range of −80 to +110 mV in 20 mV increments. Before and after these voltage pulses, the membrane was clamped to the holding voltage of −60 mV. The total time interval between voltage pulses at holding voltage was 4 s. (**B**) Normalized conductance–voltage plots ($G/G_{max}(V)$) determined for the LjTPC1 currents shown in (**A**) in the absence and presence of luminal Ca²⁺ as indicated. Best fits of the $G/V$ plots to a double Boltzmann function are given by the solid lines. Experiments were performed under symmetric K⁺ conditions (150 mM) with 1 mM cytosolic Ca²⁺ and luminal Ca²⁺ at indicated concentration. For more details on the voltage protocol and solutions, see Materials and methods. Source data are given in *Appendix 1—figure 1—source data 1*.

The online version of this article includes the following source data for appendix 1—figure 1:

**Appendix 1—figure 1—source data 1.** Quantification of voltage-clamp data shown in *Appendix 1—figure 1*.

