## [Editor Report]

Plant intracellular ion channels are poorly understood. In this manuscript, patch-clamp is used to define functional differences between two cation channels present in the vacuole. The authors present valuable findings that indicated a calcium-biding site is responsible for increased excitability in the vacuole of the faba bean plant. The experimental evidence presented is convincing and findings have practical implications for the subfield of plant electrophysiology.

---

## [Decision Letter]

**Decision letter after peer review:**

Thank you for submitting your article "Vicia faba SV channel VfTPC1 is a hyperexcitable variant of plant vacuole Two Pore Channels" for consideration by *eLife*. Your article has been reviewed by 3 peer reviewers, including Leon D Islas as the Reviewing Editor and Reviewer #1, and the evaluation has been overseen by Detlef Weigel as the Senior Editor. The following individual involved in review of your submission has agreed to reveal their identity: Vivek Garg (Reviewer #2).

Essential revisions:

After consultation, the reviewers agree that the following experiments are needed:

1. Reverse mutations in VfTPC1 need to be made and tested.

2. Expanding the single channel data to include the effect of Ca^2+^ on open probability would help provide crucial insight into channel gating (especially since max conductance is not reached for all the channel's macroscopic current recordings).

In addition, please attend to the reviewer's individual comments, which are aimed to improve the clarity of the data presentation and the manuscript's conclusions.

*Reviewer #1 (Recommendations for the authors):*

1) The order of presentation of findings in the text and the order of the figures is confusing. It jumps back and forth. The figures should be presented in the same order as they are refered to in the text.

2) Figure 1. The recording of VfTPC1 in 0 Ca^2+^ is puzzling, the tail currents are much slower than any other channels studied in the same condition and do not correspond to the findings in Figure 3 (High Ca^2+^ slows the tail current). Might this be an artifact of a poorly clamped vacuole with currents too big?

3) Figure 4 shows the effects of Ca^2+^ on the G-V curves for all channels, but, since the maximum conductance is not reached, it is impossible to understand to what extent the effect of ca^2+^ is on the V1/2 of activation or a reduction of the maximum conductance, indicating possible block by such high concentrations of Ca^2+^. This possible block of ca^2+^ should be explored in single-channel recordings.

4) Line 177. The authors state that VfTPC1 channels activate at the resting potential of the vacuole but no indication is given of this value.

5) The model employed for generation of vacuole action potentials is presented in a manner that is too simplified. More details on the equations and numerical methods are needed to enhance reproducibility.

6) Figure 4- Supplement 1. The tail currents of LjTPC1 channels are extremely slow. The current density indicates that the current magnitude is extremely large and the tail current shows signs of poor space clamp. A meaningful comparison between conditions (0 and 50 mM calcium) should be made with smaller currents.

7) Structural Figures 6 and its supplement lack detail and spatial orientation to be useful. These figures should be remade in a clearer way for the reader to get the message.

8) Tail currents in Figure 7 show a reduction of their magnitude in the presence of Ca^2+^. The authors should explain if this is explained by reduced permeability to ca^2+^ ions as compared to Na^+^ and K^+^ or if it reflects a pore blocking effect of luminal Ca^2+^.

9) The single channel data presented in Figure 8 is interesting, but it lacks context. What is the Ca^2+^ concentration used? An experiment showing the effects of Ca^2+^ on single-channel behavior would be more informative (see comment 3 above).

*Reviewer #2 (Recommendations for the authors):*

Line 246-252, authors acknowledge that E457N mutation does not add further to voltage and Ca^2+^-sensitivity of the double mutant (E605A/D606N). But the abstract and the discussion give the impression that three polymorphic residues are responsible for the above sensitivity. This needs to be clarified or the text might need some rewording to reconcile these statements.

To enhance clarity, the position of the three Ca^2+^ sensor sites and the two or three residues mediating luminal Ca^2+^ sensitivity can be depicted in a cartoon or topology model of the channel in one of the figure.

Voltage protocol should be clearly defined for each figure along with the pulse interval (either in the legend or methods) e.g. the voltage used for recording tail currents in Figure 1 is presumably the same as the holding potential. Since the deactivation is significantly different between AtTPC1 and VfTPC1, pulse interval can significantly modify the current activation kinetics.

In the previous publication authors already tried some of the individual single point mutations for some of the polymorphic residues presented in this paper. It will be worthwhile to present that data in this paper for comparison (maybe as supplementary figures).

E457 residue (site 2) is far away from the E605/D606 residues (site 3) in the model. This mutant was not tested before. It will be worth checking the current phenotype of the single mutant, E457N AtTPC1, and comparing it with the double mutant E605A/D606N.

When comparing AtTPC1 and VfTPC1 channels, I wonder if making the reverse or corresponding mutations in VfTPC1 (i.e. A607E/N608D) will result in a channel with Ca^2+^-sensitivity similar to AtTPC1. Is it possible that the luminal Ca^2+^ and the allosteric voltage sensor coupling mechanism be different in At versus Vf?

*Reviewer #3 (Recommendations for the authors):*

My specific comments are as follows.

1. The physiological luminal Ca^2+^ concentrations of plant vacuoles are between 1.5 and 2.3 mM, yet the lowest Ca^2+^ concentration used in this study was 10 mM. Are there any measurable differences between AtTPC1 and VfTPC1 at physiological ca^2+^ concentrations? Is there a plant growth phenotype when the Ca^2+^ sensors of AtTPC1 are mutated?

2. In Figures 1 and 9, vacuoles from Arabidopsis tpc1-2 should be included as negative controls.

3. For Figure 2, the activation is both time and voltage dependent. It might be helpful to compare the time courses of the close state vs. the fully activated state. According to the G-V curve in Figure 4, AtTPC1 is not fully activated at 90 mV. Hence, a stronger depolarization might be necessary for the demo trace in Figure 2A. Likewise, the deactivation in Figure 3 should also be studied with the fully activated AtTPC1. In Figure 3A, it is not clear which trace is for +80 mV or +100 mV.

4. The authors suggested that three luminal residues are responsible for the distinct properties of TPC1 from different species. While it was shown that mutating the residues in AtTPC1 into those in VfTPC1 made the AtTPC1 channel resembling VfTPC1, how about the reversed mutations in VfTPC1? Do they make the VfTPC1 channel resembling AtTPC1?

5. The deactivation of TPC1 from *Lotus japonicus* appeared to be much slower than that of VfTPC1. Given these residues (i.e, A607 and N608) are the same for both channels, it is unlikely they are the primary determinants responsible for the slow deactivation.

6. In Figure 7A, the scale bars should be numbered, and the controls from non-transfected HEK293 cells should be shown. In addition, the selectivity data from AtTPC1, AtTPC1-E605A/D606N, and VfTPC1-A607E/N608D in HEK293 cells should be compared.

7. For E605A/D606N and E457N/E605A/D606N mutant channels, longer depolarizations might be needed to see the steady state currents in 50 mM ca^2+^.

8. In Figure 8A, the genetic background should be mentioned in the legend. Additionally, single channel recordings of non-transfected and VfTPC1-A607E/N608D -transfected cells should be provided.

9. Indeed, all the experiment conditions should be clearly described in the figure legend, e.g., the steady state currents in Figure 1 and the genetic background of experiments in Figure 2, Figure 3, and Figure 9.

10. What is the Ca^2+^ dose-dependence of the inhibition of TPC1 open probability?

---

## [Author Response]

Essential revisions:After consultation, the reviewers agree that the following experiments are needed:1. Reverse mutations in VfTPC1 need to be made and tested.

As requested, we tested the effect of reverse mutations (N458E/A607E/N608D) in VfTPC1 on its response to membrane voltage and luminal Ca^2+^. The results are described below in the response to the last comment of reviewer 2.

2. Expanding the single channel data to include the effect of Ca^2+^ on open probability would help provide crucial insight into channel gating (especially since max conductance is not reached for all the channel's macroscopic current recordings).

Our data excludes the possibility that the current reduction is predominantly caused by a Ca^2+^ block. For this reason, and in view of published data on the TPC1 single channel conductance and open probability under Ca^2+^ loads, we did not examine the luminal Ca^2+^ effect at the single channel level. Please see our detailed response to comment 3 of reviewer 1.

In addition, please attend to the reviewer's individual comments, which are aimed to improve the clarity of the data presentation and the manuscript's conclusions.

We have addressed each of the reviewer's comments, please see details below.

Reviewer #1 (Recommendations for the authors):1) The order of presentation of findings in the text and the order of the figures is confusing. It jumps back and forth. The figures should be presented in the same order as they are refered to in the text.

We have tried to follow *eLife*’s policy for supplementary illustrations which is to link the supplemental figures to the main figures. However, as the previous ordering strategy seems to be difficult for the readership to read in some places in our manuscript, we have separated some figure supplements from the main figures and now show some of them as main figures or as Appendix figure in the order as they are referred in the main text. Accordingly, the numbering of the figures has been changed.

2) Figure 1. The recording of vfTPC1 in 0 Ca^2+^ is puzzling, the tail currents are much slower than any other channels studied in the same condition and do not correspond to the findings in Figure 3 (High Ca^2+^ slows the tail current). Might this be an artifact of a poorly clamped vacuole with currents too big?

We have no reason to believe that the slower relaxation of the VfTPC1 tail currents compared with the AtTPC1 channel variants in Figure 1A (now labeled as Figure 2A) is an artifact due to poor clamping of the vacuole. This is supported by the mean half-deactivation times in Figure 3B (now labeled as Figure 4B): The mean half-deactivation times of VfTPC1 derived from current recordings at 0 mM Ca^2+^ are significantly slower than those of the AtTPC1 wild type channel at higher negative membrane voltages (Figure 3B; now labeled as Figure 4B) and becomes faster with an increase to 10 mM Ca^2+^ (Figure 3B, C; now labeled as Figure 4B, C). These experiments were characterized by similar patch clamp parameters such as membrane capacitance (*C_m_*), series resistance (*R_series_*) and absolute current amplitudes (*I*) at the pre-activating voltage pulse applied before the deactivating voltage pulse, as given by the following mean values ± SE: AtTPC1-WT (n=5) *C_m_* = 42.7 ± 1.9 pF, *R_series_* = 2.6 ± 0.3 MΩ, *I* = 8.7 ± 1.6 nA; AtTPC1-D605A/E606N (n=4) *C_m_* = 47.0 ± 4.2 pF, *R_series_* = 2.3 ± 0.2 MΩ, *I* = 11.0 ± 1.1 nA; AtTPC1-D605A/E606N/E457N (n=5) *C_m_* = 48.0 ± 2.0 pF, *R_series_* = 2.7 ± 0.2 MΩ, *I* = 8.8 ± 0.6 nA; VtTPC1 (n=4) *C_m_* = 45.5 ± 4.3 pF, *R_series_* = 2.9 ± 0.2 MΩ, *I* = 10.3 ± 1.0 nA. In comparison, AtTPC1-WT, the double and triple AtTPC1 mutants (AtTPC1-D605A/E606N, AtTPC1-D605A/E606N/ E457N) show comparable deactivation kinetics. We would also like to point out that such slow deactivation kinetics as found here for VfTPC1 is not uncommon among the TPC1 channels of other plant species. For example, the TPC1 channel variant from the moss *Physcomitrella patens* (PpTPC1a) also shows much slower deactivation than AtTPC1 (Dadacz-Narloch et al. 2011, DOI: 10.1105/tpc.111.086751). See also the response to comment 6 (reviewer 1) and to comment 5 of reviewer 3. In the revised manuscript, we now go into more detail about the different deactivation behavior.

3) Figure 4 shows the effects of Ca^2+^ on the G-V curves for all channels, but, since the maximum conductance is not reached, it is impossible to understand to what extent the effect of Ca^2+^ is on the V1/2 of activation or a reduction of the maximum conductance, indicating possible block by such high concentrations of Ca^2+^. This possible block of ca^2+^ should be explored in single-channel recordings.

The Ca^2+^-induced shift of the activation threshold shown in Figure 4 and also in Figure 1A (now labeled as Figure 5 and Figure 2A, respectively) clearly excludes the possibility that the current reduction predominantly results from a block of Ca^2+^. Figure 4A (now labeled as Figure 5A) further shows that in 0 Ca^2+^ the G/V curves almost reached their maximum, in particular for VfTPC1, the AtTPC1 double and triple mutant. Thus, the slope of the curve at the midpoint activation voltage is rather well determined. With increasing Ca^2+^, this slope does not change markedly, which supports the notion that the dominant effect is on the midpoint activation voltage V_1/2_ and not on G_max_. Based on these considerations, we refrained from exhaustive single channel measurements to additionally disprove a fact that is rather unlikely. Nevertheless, a Ca^2+^ block might contribute to the observed current reduction. In current recordings on *Vicia faba* vacuoles, Ward and Schroeder (1994, see below Author response table 1) showed that the permeant cations K^+^ and Ca^2+^ both affect the single channel conductance. With only Ca^2+^ as a permeant cation on both sides (50_vac_/5_cyt_ mM), the single channel conductance of VfTPC1 was only 16 pS, but increased to 155 pS when K^+^ (100 mM) was the major cation on both sides of the vacuole membrane. Under symmetric K-based solute conditions (100 mM) on both sides of the vacuole membrane, the presence of 5 mM Ca^2+^ at the luminal side caused a decrease in the single channel conductance from 250 pS (see below Author response table 2; our recordings, 0 mM luminal Ca^2+^) to 155 pS (Ward and Schroeder 1994). Thus, luminal Ca^2+^ at a concentration of 10 or 50 mM under symmetric K conditions (150 mM) in our whole-vacuole current recordings probably affected not only the TPC1 voltage dependence but also the single channel conductance even though the negative Ca^2+^ effect on the ion transport capacity may be attenuated by the higher K^+^ concentration (150 vs 100 mM K). Note that, the Ca^2+^ effect on the ion transport capacity does not appear to correlate with Ca^2+^ sensor site 3, as this side is not functional in VfTPC1.

**Author response table 1. sa2table1:** Single channel conductance of TPC1/SV currents in *Vicia faba* guard cells (Ward and Schroeder 1995, doi 10.1105/tpc.6.5.669).

Pipette solution in mM(luminal side)	Bath medium in mM(cytosolic side)	Single channel conductance
**K** ^ **+** ^	**Ca** ^ **2+** ^	**pH**	**K** ^ **+** ^	**Ca** ^ **2+** ^	**pH**	
0	50	5.5	0	5	7.2	16 pS
0	50	5.5	100	5	7.2	117 pS
100	5	5.5	100	1	7.2	155 pS

**Author response table 2. sa2table2:** Single channel conductance of VfTPC1 expressed in *Arabidopsis thaliana* vacuoles (Lu al., present manuscript, Figure 11).

Pipette solution in mM(luminal side)	Bath medium in mM(cytosolic side)	Single channel conductance
**K** ^ **+** ^	**Ca** ^ **2+** ^	**pH**	**K** ^ **+** ^	**Ca** ^ **2+** ^	**pH**	
100	0	5.5	100	0.5	7.5	250 pS

In addition, our group has studied this question also on *Arabidopsis thaliana* wild type and *fou2* mutant plants in Beyhl et al. (2009, DOI: 10.1111/j.1365313X.2009.03820.x). It was demonstrated that a luminal Ca^2+^ concentration of 0.1 and 1 mM under symmetric 100 mM K^+^ only affected the single channel open probability and did not reduce the single current amplitude of both wild type and *fou2* TPC1 channels (Beyhl et al. 2009, Figure 5), the latter being a mimic of our AtTPC1-D605A/E606N mutant.

4) Line 177. The authors state that VfTPC1 channels activate at the resting potential of the vacuole but no indication is given of this value.

The reviewer is right that the resting potential of the vacuole should be mentioned. We revised the manuscript accordingly in the introduction and discussion.

5) The model employed for generation of vacuole action potentials is presented in a manner that is too simplified. More details on the equations and numerical methods are needed to enhance reproducibility.

We provided now further details on the *in silico* experiments (see Supplementary File 3). Although they have already been described in our cited earlier paper Jaslan et al. 2019 (doi: 10.1038/s41467-019-10599-x.), for the convenience of the reader this additional information is now provided in the revised manuscript.

6) Figure 4- Supplement 1. The tail currents of LjTPC1 channels are extremely slow. The current density indicates that the current magnitude is extremely large and the tail current shows signs of poor space clamp. A meaningful comparison between conditions (0 and 50 mM calcium) should be made with smaller currents.

Thank you for the critical comment. We agree that space clamp problems cannot be fully excluded for this experiment with LjTPC1 channels shown under 0 mM luminal Ca^2+^ because the membrane currents were very large indeed. After re-inspection of our LjTPC1 experiments under 0 mM Ca^2+^, we now found an experiment with lower TPC1 currents at +110 mV. This current amplitude was comparable to those recorded at +110 mV with the other TPC1 channel variants (VfTPC1: 595±10 pA/pF (n=5); AtTPC1-WT 571±66 pA/pF (n=5); AtTPC1-D605AD606N 671±81 pA/pF (n=5); mean±SE). Nevertheless, the tail currents were still very slow as illustrated in the relabeled Appendix 1—figure 1. However, such extremely slow relaxation for TPC1 channel is not unusual. The At2HsTPC2 mutant, in which three selectivity filter residues in AtTPC1 were replaced by those of human HsTPC2, also showed very slow tail currents (Guo et al. 2017, Figure S2, DOI: 10.1073/pnas.1616191114). In addition, see also response to comment 5 of reviewer 3.

7) Structural Figures 6 and its supplement lack detail and spatial orientation to be useful. These figures should be remade in a clearer way for the reader to get the message.

We agree that the spatial orientations were lacking and rewrote the figure caption and reference to the supplementary material. We feel that this does define the orientations much more precisely and coherently. Please note that the former Figure 6 and its figure supplement was relabeled as Figure 9.

In addition, also considering comment 2 of reviewer 2, we revised Figure 6—figure supplement 1 (now labeled as Figure 9—figure supplement 1) as follows:

We add a topology cartoon of the VfTPC1 protein showing positions of the three luminal Ca^2+^ bindings sites in comparison to Arabidopsis AtTPC1.The amino acid residues of the three luminal Ca^2+^ bindings sites are highlighted in color in the 3D model of the VfTPC1 channel.We now show not only a front view of the 3D VfTPC1 model, but also an orthogonal top view.

8) Tail currents in Figure 7 show a reduction of their magnitude in the presence of Ca^2+^. The authors should explain if this is explained by reduced permeability to Ca^2+^ ions as compared to Na^+^ and K^+^ or if it reflects a pore blocking effect of luminal ca^2+^.

For clarity, we have slightly revised Figure 7 (now labeled as Figure 10) and show now the concentration of the relevant permeant cations on both sides of the HEK cell plasma membrane above the raw current traces in Figure 7C-E (now labeled as Figure 10C-E). Additionally, the dashed zero lines of the current responses in C-E were set to the same level, and the currents of nontransfected HEK293 cells are shown as a negative control for all three solute conditions.

Yes, the tail currents recorded in Figure 7C-E (now labeled as Figure 10C-E) from the same HEK cell were reduced under extracellular Ca^2+^ condition compared with K^+^ or Na^+^. On the one hand the lower tail current amplitude in Ca^2+^ can be attributed to the 10-fold lower Ca^2+^ concentration (15 mM) compared with Na^+^ or K^+^ (each 150 mM). On the other hand, it also is in line with the recent mechanistic explanation that the relative permeability of TPC1 (P_Ca_/P_Na_ » 5:1) does not correlate with the actual Ca^2+^ conductivity (Navarro-Retamal et al., 2021). Or in other words, Ca^2+^ ions cannot permeate the channel better than K^+^ or Na^+^ ion which may additionally lead to a lower tail current amplitude. This is further supported by the very small single channel conductance under Ca^2+^based solute conditions compared to K^+^-based ones. Furthermore, the single channel conductance is reduced at a high luminal Ca^2+^ addition under K^+^-based solute conditions (see above the author response Tables 1 and 2 given in response to comment 3 from reviewer 1). This point is now addressed in the legend to Figure 7 (now labeled as Figure 10).

9) The single channel data presented in Figure 8 is interesting, but it lacks context. What is the Ca^2+^ concentration used? An experiment showing the effects of Ca^2+^ on single-channel behavior would be more informative (see comment 3 above).

In Figure 8 (now labeled as Figure 11), the single channel recordings were performed using the following K-based solutions: The bath medium contained 100 mM KCl, 0.5 mM CaCl_2_, 10 mM Hepes (pH 7.5/Tris), and the pipette medium consisted of 100 mM KCl, 2 mM MgCl2, 2 mM EGTA, 10 mM MES (pH 5.5/Tris). Although this solute composition was mentioned in the Material and Methods under the section “patch clamp experiments with membrane patches”, for clarity and ease of comparison, the concentration of the relevant cations K^+^ and Ca^2+^ used for these recordings is now additionally given in the legend of Figure 8 (now labeled as Figure 11). Regarding the Ca^2+^ effect on the single channel behavior, please see the response to comment 3 from reviewer 1 above.

Reviewer #2 (Recommendations for the authors):Line 246-252, authors acknowledge that E457N mutation does not add further to voltage and Ca^2+^-sensitivity of the double mutant (E605A/D606N). But the abstract and the discussion give the impression that three polymorphic residues are responsible for the above sensitivity. This needs to be clarified or the text might need some rewording to reconcile these statements.

We agree and revised the abstract accordingly. In the first paragraph of the discussion, it was already stated that “The dual polymorphism of two residues within the Ca^2+^ sensor site 3 in the luminal pore entrance is most likely responsible for this gating behavior of VfTPC1, which is clearly distinct from the AtTPC1 channel of Brassicaceae.”

To enhance clarity, the position of the three Ca^2+^ sensor sites and the two or three residues mediating luminal Ca^2+^ sensitivity can be depicted in a cartoon or topology model of the channel in one of the figure.

We revised the Figure 6—figure supplement 1 (now labeled as Figure 9—figure supplement 1) accordingly. For further details, please see our response to comment 7 of reviewer 1.

Voltage protocol should be clearly defined for each figure along with the pulse interval (either in the legend or methods) e.g. the voltage used for recording tail currents in Figure 1 is presumably the same as the holding potential. Since the deactivation is significantly different between AtTPC1 and VfTPC1, pulse interval can significantly modify the current activation kinetics.

We agree and revised the relevant Figure legends and sections in Materials and methods accordingly. Since the time interval between the activation and deactivation voltage pulses at holding voltage (-60 mV) were very long (4 s and 4.3 s, respectively), the pulse interval should not have affected the kinetics.

In the previous publication authors already tried some of the individual single point mutations for some of the polymorphic residues presented in this paper. It will be worthwhile to present that data in this paper for comparison (maybe as supplementary figures).

With due respect, we believe that the citation of our previous paper (Dickinson et al. 2021, DOI: 10.1073/pnas.2110936119) presenting single point mutations for some of the polymorphic residues in the ca^2+^ sensor site 3 is sufficient. We have mentioned in the present manuscript that AtTPC1 wild-type data at 0 and 10 mM Ca^2+^ are identical to those shown in Dickinson et al., 2021. Therefore, it is easy to compare the results of both papers. Otherwise, we believe it will be too redundant to show already published data in a supplementary figure again.

E457 residue (site 2) is far away from the E605/D606 residues (site 3) in the model. This mutant was not tested before. It will be worth checking the current phenotype of the single mutant, E457N AtTPC1, and comparing it with the double mutant E605A/D606N.

We have already tested the single mutants AtTPC1-E457Q and AtTPC1-E457N in a previous paper under identical ionic patch clamp solutions (Dadacz-Narloch et al. 2011, doi: 10.1105/tpc.111.086751). The luminal Ca^2+^ sensitivity of these mutants was significantly reduced compared with AtTPC1 wild type, but without promoting voltage-dependent activation, as half-activation potentials (V_1_, V_2_) were shifted to more positive values and activation kinetics were slowed down. At Ca^2+^ sensor site 2, VfTPC1 now contains N458 instead E458 at the homologous site to AtTPC1-E457. With respect to the effect of E457 on the luminal Ca^2+^ sensitivity of AtTPC1, therefore we further examined whether a neutral residue at site 457 (E457N) intensifies the effects of the double mutations in Ca^2+^ sensor site 3 (E605A/D606N) on AtTPC1 channel gating under different luminal Ca^2+^ concentrations. As mentioned in the previous manuscript version at the end of the result section “Polymorphic AtTPC1 triple mutant mimics the VfTPC1 channel features”, we found and concluded the following:

“An additional E457N replacement did not further affect the transition from an Arabidopsis-like to a VfTPC1-like gating behavior. Considering the attenuating effect of neutralized site 457 (E → N/Q) on voltage-dependent activation of AtTPC1 (Dadacz-Narloch et al., 2011), the hyperactivity of the native *Vicia faba* TPC1 channel is most likely related to the altered Ca^2+^ sensor site 3.” Since the reader, like the reviewer, may overlook the fact that we have already addressed this AtTPC1-E457 issue, we have revised this section as followed: “Thus, the E605A/D606N exchange is already sufficient to provide AtTPC1 with the voltage and Ca^2+^ sensitivity of VfTPC1. The additional E457N replacement in the AtTPC1 E605A/D606N mutant background, however, did not further affect the transition from an Arabidopsis-like to a VfTPC1-like gating behavior, nor did it further reduce luminal Ca^2+^ sensitivity. Dadacz-Narloch et al. (2011) reported that neutral residues at site 457 (E → N/Q) in AtTPC1 made the channel significantly less sensitive to luminal Ca^2+^ but attenuated voltage dependent activation compared to wild type channels, as midpoint voltages (V_1_, V_2_) were shifted to more positive values. Thus, the hyperactivity of the native *Vicia faba* TPC1 channel is most likely related to the altered Ca^2+^ sensor site 3 rather than to Ca^2+^ sensor site 2. Considering further the similar luminal Ca^2+^ sensitivity of E605A/D606N and E457N/E605A/D606N (Figure 5C), the two Ca^2+^ sensor sites 2 and 3 with their residues E457 and E605/D606, respectively, do not have additive effects on channel gating.

When comparing AtTPC1 and VfTPC1 channels, I wonder if making the reverse or corresponding mutations in VfTPC1 (i.e. A607E/N608D) will result in a channel with Ca^2+^-sensitivity similar to AtTPC1. Is it possible that the luminal Ca^2+^ and the allosteric voltage sensor coupling mechanism be different in At versus Vf?

Using the VfTPC1 triple mutant (N458E/A607E/N608D), we tested whether such inverse mutations cause VfTPC1 to become AtTPC1-like. Unfortunately, the gating behavior of this VfTPC1 triple mutant was only slightly different from VfTPC1 wild type under 0 and 10 mM luminal Ca^2+^. The triple mutant still started to activate close to the vacuolar resting membrane voltage around -30 mV and similarly responded to an increase to luminal Ca^2+^ (10 mM) (see Figure 7). Thus, the VfTPC1 triple mutant did not gain the high luminal Ca^2+^ sensitivity and high depolarization-dependent gating behavior of AtTPC1, indicating that other additional structural adaptations have evolved and counteract these “gain-of-function” efforts. The literature shows that in contrast to “loss-of-function” mutations (successfully applied here to AtTPC1), such reverse gain-of-function approaches with plant ion channels also failed before (Michard et al. 2005, doi.org/10.1085/jgp.200509413; Johansson et al. 2006, doi.org/10.1111/j.1365-313X.2006.02690.x). There are many subtle structural details that are important for the particular mechanism, not all of which can be identified and transplanted.

The VfTPC1 triple mutant data are now presented and discussed in the manuscript.

Reviewer #3 (Recommendations for the authors):My specific comments are as follows.1. The physiological luminal Ca^2+^ concentrations of plant vacuoles are between 1.5 and 2.3 mM, yet the lowest Ca^2+^ concentration used in this study was 10 mM. Are there any measurable differences between AtTPC1 and VfTPC1 at physiological Ca^2+^ concentrations?

We would like to point out that the physiological luminal Ca^2+^ concentrations of plant vacuoles range from as low as 0.2 mM up to 2.3 mM as described in the cited review by Schönknecht (2013, doi: 10.3390/plants2040589). Therefore, we used this lowest physiological Ca^2+^ concentration of 0.2 mM measured in higher plants in our current clamp and also voltage clamp experiments to record TPC1-dependent membrane polarization and the corresponding TPC1 currents (Figure 12, Figure 12—figure supplement 1). The results have been described in the section: “Vacuoles with VfTPC1 and AtTPC1 triple mutant are hyperexcitable”. Briefly, in these current clamp experiments the vacuoles equipped with VfTPC1 channels were hyperexcitable; they remained depolarized at a voltage of about 0 mV (i.e. at the equilibrium potential for K^+^) during the entire subsequent recording period of 10 s, regardless of the stimulus intensity. In contrast, when AtTPC1-equipped vacuoles were challenged with even the highest current stimulus (1 nA), the post-stimulus voltage remained depolarized for only a short period (*t*_plateau_ ~ 0.4 s, Figure 12—source data 1) before relaxation to the resting voltage occurred. In comparison to wild type AtTPC1, the AtTPC1-triple mutant vacuoles were also hyperexcitable, behaving very much like the VfTPC1-equipped vacuoles.

The further analysis of the corresponding TPC1 currents of these same vacuoles revealed, that at this physiological luminal Ca^2+^ concentration (0.2 mM), the voltage activation threshold of both, VfTPC1 and the AtTPC1 triple mutant E457N/E605A/D606N, but not AtTPC1 wild type was close to the resting membrane voltage of around -30 mV (Figure 12—figure supplement 1E, F). This explains the hyperexcitable vacuoles when equipped with VfTPC1 and the AtTPC1 triple mutant E457N/E605A/D606N.

Is there a plant growth phenotype when the Ca^2+^ sensors of AtTPC1 are mutated?

Yes, as mentioned in the introduction (third paragraph), there is a phenotype when the Ca^2+^ sensor site 1 was destroyed by the mutation at site 454 in AtTPC1 (D454N). This so-called *fou2* mutant shows a retarded growth phenotype, appears to be chronically wounded and therefore produces large amounts of the wounding hormone jasmonate (Bonaventure et al., 2007, DOI: 10.1111/j.1365-313X.2006.03002.x ). Since the *fou2* TPC1 channel behaves like a hyperactive TPC1 channel, the *fou2*-equipped vacuoles are hyperexcitable (Jaslan et al. 2019, doi: 10.1038/s41467-019-10599-x) similar to vacuoles equipped with VfTPC1 or the AtTPC1 triple mutant E457N/E505A/D606N (Figure 12). According to the resolved 3D structure of the *fou2* channel (Dickinson et al. 2021, DOI: 10.1073/pnas.2110936119 ) and our VfTPC1 model in this paper, the key residues E605, D606 of the Ca^2+^ sensor site 3 are now rearranged, preventing binding of luminal Ca^2+^, as electrophysiologically observed for the AtTPC1 mutants E505A/D606N and E457N/E505A/D606N. Therefore, we would expect that AtTPC1 mutant plants E505A/D606N show a similar growth phenotype as *fou2*.

2. In Figures 1 and 9, vacuoles from Arabidopsis tpc1-2 should be included as negative controls.

Such control experiments with *attpc1-2* vacuoles for Figures 1 and 9 (now labeled as Figures 2 and 12) have been already presented in previous publications of our group such as Jaslan et al. 2016 (doi: 10.1111/plb.12478) and Jaslan et al. 2019 (doi: 10.1038/s41467-019-10599-x), respectively. Since all these experiments were performed under identical ionic conditions, we feel that the manuscript will not gain much in showing published negative control experiments once again. However, we understand the reviewer's concerns very well and certainly agree that negative controls are important. Therefore, to convince the reader about the quality of our electrophysiological data, we now mention in Material/Methods that negative controls have been published in these papers (Jaslan et al. 2016, 2019).

3. For Figure 2, the activation is both time and voltage dependent. It might be helpful to compare the time courses of the close state vs. the fully activated state. According to the G-V curve in Figure 4, AtTPC1 is not fully activated at 90 mV. Hence, a stronger depolarization might be necessary for the demo trace in Figure 2A. Likewise, the deactivation in Figure 3 should also be studied with the fully activated AtTPC1. In Figure 3A, it is not clear which trace is for +80 mV or +100 mV.

In order to ensure that the TPC1 channels are fully closed, the vacuole membrane was clamped for 4.3 s to the (channel-closing) holding voltage of -60 mV between the applied voltage pulses. Therefore, activation of the TPC1 channels elicited upon increasing depolarized membrane voltage pulses was always achieved from the fully closed channel state. In Figure 2A (now labeled as 3A), the normalized demo traces recorded at +90 mV illustrate the significant differences in activation kinetics of the different channel variants found at less and higher depolarized voltages as well (Figure 2B-D; now labeled as Figure 3B-D). Given the presented data set, we do we do not think the reader gains more by displaying demo traces at different voltages.

The legend of Figure 3A (now labeled as 4A) now indicates that the pre-pulse voltage of +100 and +80 mV was used for activation of the three AtTPC1 channel variants and VfTPC1, respectively, to study the deactivation process. Unfortunately, it was not possible to pre-activate at +100 mV the VfTPC1 channels in the same way as the AtTPC1 channel variants, since the vacuole got lost. A pre-activating voltage pulse of +100 mV combined with the multiple subsequent deactivating voltage pulses was obviously too stressful for the VfTPC1-equipped vacuoles. However, because of the negatively shifted relative open-channel probability of VfTPC1, deactivation of VfTPC1 was examined as for the other channel types with mostly “fully open” channels (rel. P_o_
> 0.85 at +80 and +100 mV; see G/V curves in Figure 5A). In addition, the deactivation kinetics at different membrane voltages were always investigated with a constant number of open TPC1 channels in each experiment by always pre-activating at the same prepulse voltage of either +80 or +100 mV.

4. The authors suggested that three luminal residues are responsible for the distinct properties of TPC1 from different species. While it was shown that mutating the residues in AtTPC1 into those in VfTPC1 made the AtTPC1 channel resembling VfTPC1, how about the reversed mutations in VfTPC1? Do they make the VfTPC1 channel resembling AtTPC1?

Using the VfTPC1 triple mutant N458E/A607E/N608D, we tested whether the VfTPC1 features become AtTPC1-like. Unfortunately, the gating behavior of this VfTPC1 triple mutant was only slightly different from VfTPC1 wild type under 0 and 10 mM Ca^2+^, as described in more detail in our answer to the last comment from reviewer 2.

5. The deactivation of TPC1 from *Lotus japonicus* appeared to be much slower than that of VfTPC1. Given these residues (i.e, A607 and N608) are the same for both channels, it is unlikely they are the primary determinants responsible for the slow deactivation.

We absolutely agree with the conclusion of the reviewer. This is also supported by the PpTPC1a channel variant from the moss *Physcomitrella patens.* The PpTPC1a channel shows much slower deactivation than AtTPC1 (Dadacz-Narloch et al. 2011, DOI: 10.1105/tpc.111.086751). Nevertheless, like AtTPC1, PpTPC1a harbors also functional Ca^2+^ sensor sites 1, 2 and 3 (see Figure 8 in the revised manuscript), as indicated by the AtTPC1-like luminal ca^2+^ sensitivity and voltage dependence of PpTPC1a (Dadacz-Narloch et al. 2011, DOI: 10.1105/tpc.111.086751). Additionally, in contrast to VfTPC1, the AtTPC1-mutant E6045E/D606N shows deactivation kinetics similar to AtTPC1 wild type while its Ca^2+^ sensitivity and voltage dependence are VfTPC1-like (Figures 4, 5). Thus, we can indeed conclude that other functional domains than the Ca^2+^ sensor site 3 determines the deactivation behavior of TPC1 channels. We now address this point in the result section “Polymorphic AtTPC1 triple mutant mimics the VfTPC1 channel features” in the manuscript.

Note that the single mutation of Ca^2+^ sensor 3 in PpTPC1a (ESD) compared to AtTPC1 (EDD) will not affect the gating behavior and Ca^2+^ sensitivity as shown in Dickinson et al. 2021 (DOI: 10.1073/pnas.2110936119). This is in line with the AtTPC1-like luminal Ca^2+^ sensitivity and voltage dependence of PpTPC1a (Dadacz-Narloch et al. 2011, DOI: 10.1105/tpc.111.086751).

6. In Figure 7A, the scale bars should be numbered, and the controls from non-transfected HEK293 cells should be shown. In addition, the selectivity data from AtTPC1, AtTPC1-E605A/D606N, and VfTPC1-A607E/N608D in HEK293 cells should be compared.

In Figure 7A (now labeled as Figure 10), we have revised the scale bar, and we now show also the controls from non-transfected HEK293 cells under the three different solute conditions. However, we did not examine the VfTPC1A607E/N608D mutant because (i) the reverse mutations N458E/A607E/N608D in VfTPC1 did not convert the hypersensitivity and luminal Ca^2+^ insensitivity of VfTPC1 into AtTPC1-like features (for more details, see our answer to the last comment 6 from reviewer 2), and (ii) the relative permeability ratios of AtTPC1 wild type and the VfTPC1-like AtTPC1 mutant E605A/D606A were very similar.

7. For E605A/D606N and E457N/E605A/D606N mutant channels, longer depolarizations might be needed to see the steady state currents in 50 mM ca^2+^.

We agree that the depolarization-induced TPC1 currents in the presence of 50 mM Ca^2+^ are in the quasi-steady state and thus very close to reaching the steady-state current level. We would also have liked to extend the activating voltage pulses further, but already the applied 1-s lasting voltage pulses were quite stressful for the patched vacuole. Therefore, we have applied a very long time-interval of 4 s between the voltage pulses at the holding voltage, so that the vacuoles could recover, and the currents could relax fully at holding voltage. Or in other words, the risk of losing the vacuole at an even longer voltage pulse time would greatly increase; therefore, the 1-s-lasting voltage pulses for TPC1 current activation was already a compromise between lifetime of the patched vacuole and the achievement of the steady-state current level. However, since we are near the steady-state current level, we feel our results to be quite solid already.

8. In Figure 8A, the genetic background should be mentioned in the legend. Additionally, single channel recordings of non-transfected and VfTPC1-A607E/N608D -transfected cells should be provided.

We have revised the legend of Figure 8 (now labeled as Figure 11), accordingly. However, we did not additionally examine the single channel conductance of the VfTPC1-A607E/N608D mutant because (i) the reverse mutations N458E/A607E/N608D in VfTPC1 did not convert the hypersensitivity and luminal Ca^2+^ insensitivity of VfTPC1 into AtTPC1-like features (for more details, see our answer to the last comment from reviewer 2) and (ii) the single channel conductance of the AtTPC1 wild type and the VfTPC1-like AtTPC1 mutant E605A/D606A were very similar as well.

9. Indeed, all the experiment conditions should be clearly described in the figure legend, e.g., the steady state currents in Figure 1 and the genetic background of experiments in Figure 2, Figure 3, and Figure 9.

We agree and have revised the figure legends accordingly.

10. What is the ca^2+^ dose-dependence of the inhibition of TPC1 open probability?

In this work we have determined the midpoint activation voltages V_1_/V_2_ under three luminal Ca^2+^ conditions (0, 10, 50 mM) (Figure 5). The difference in the corresponding Ca^2+^-induced shift in the V_1_/V_2_ values points to a different Ca^2+^ dose dependency of the TPC1 channel variants tested. This can be supported when the V_1_ and V_2_ values are plotted against the corresponding Ca^2+^ concentrations (see Author response image 1). The slope in the Ca^2+^(10 mM)-induced increase in the midpoint activation voltages – particularly V_1_ – of the AtTPC1 wild type appears to be steeper than for the other three TPC1 channel variants (VfTPC1, AtTPC1-double/triple mutant). However, more Ca^2+^ concentrations below 10 mM need to be tested in the future to gain more insights into it. Therefore, we did not further address this issue in the manuscript.

**Author response image 1. sa2fig1:** Luminal Ca^2+^ dose dependence of voltage activation of TPC1 channel variants. Absolute (A) and normalized (B) voltage activation midpoints *V*_1_ and *V*_2_ derived from G/V curves (Figure 5) of the indicated TPC1 channel variants plotted against the corresponding luminal Ca^2+^ concentration.